# Human Texts Are Outliers: Detecting LLM-generated Texts via Out-of-distribution Detection

**Cong Zeng**[*]
MBZUAI

**Shengkun Tang**[*]
MBZUAI

**Yuanzhou Chen**
UCLA

**Zhiqiang Shen**
MBZUAI

**Wenchao Yu**
NEC Lab

**Xujiang Zhao**
NEC Lab

**Haifeng Chen**
NEC Lab

**Wei Cheng**[✉]
NEC Lab

**Zhiqiang Xu**[✉]
MBZUAI

## Abstract

The rapid advancement of large language models (LLMs) such as ChatGPT, DeepSeek, and Claude has significantly increased the presence of AI-generated text in digital communication. This trend has heightened the need for reliable detection methods to distinguish between human-authored and machine-generated content. Existing approaches both zero-shot methods and supervised classifiers largely conceptualize this task as a binary classification problem, often leading to poor generalization across domains and models. In this paper, we argue that such a binary formulation fundamentally mischaracterizes the detection task by assuming a coherent representation of human-written texts. In reality, human texts do not constitute a unified distribution, and their diversity cannot be effectively captured through limited sampling. This causes previous classifiers to memorize observed OOD characteristics rather than learn the essence of 'non-ID' behavior, limiting generalization to unseen human-authored inputs. Based on this observation, we propose reframing the detection task as an out-of-distribution (OOD) detection problem, treating human-written texts as distributional outliers while machine-generated texts are in-distribution (ID) samples. To this end, we develop a detection framework using one-class learning method including DeepSVDD and HRN, and score-based learning techniques such as energy-based method, enabling robust and generalizable performance. Extensive experiments across multiple datasets validate the effectiveness of our OOD-based approach. Specifically, the OOD-based method achieves 98.3% AUROC and AUPR with only 8.9% FPR95 on DeepFake dataset. Moreover, we test our detection framework on multilingual, attacked, and unseen-model and -domain text settings, demonstrating the robustness and generalizability of our framework. Code, pretrained weights, and demo will be released openly at https://github.com/cong-zeng/ood-llm-detect.

## 1 Introduction

In recent years, with the continuous iteration and rapid advancement of large language models (LLMs) such as ChatGPT [1], DeepSeek [2] and Claude [3], the adoption of LLM-powered products in everyday life and professional settings has increased significantly [4, 5, 6]. While these text-generation models have greatly enhanced productivity and efficiency, they also pose risks of misuse [7, 8, 9]. The phenomenon of hallucinations, instances where LLMs generate incorrect or misleading content can lead to serious consequences if such outputs are inadvertently relied upon [10]. As LLMs become increasingly powerful, there is a critical need for detection techniques that are stable, robust, and accurate in distinguishing machine-generated text from human-written content[11, 12].

---

[*]Equal contribution. The names are ordered by coin flip. [✉]Corresponding authors.

39th Conference on Neural Information Processing Systems (NeurIPS 2025).

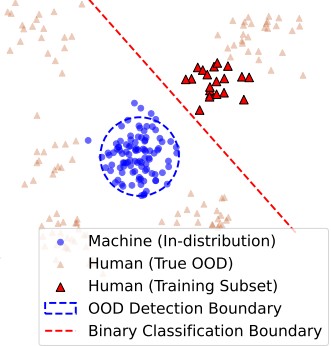

| Data Type | Metric | Distance |
|---|---|---|
| Clean Data | Intra-distance LLM | 0.3014 |
| | Intra-distance human | 0.4747 |
| | Inter-distance | 1.6048 |
| Attacked Data | Intra-distance LLM | 0.1929 |
| | Intra-distance human | 0.8698 |
| | Inter-distance | 0.7859 |

**Machine (In-distribution)**
**Human (True OOD)**
**Human (Training Subset)**
**OOD Detection Boundary**
**Binary Classification Boundary**

Figure 1: **Decision boundaries and model performance comparison.** Left: Decision boundaries under distributional asymmetry. The binary classifier separates machine-generated text from limited human-written samples but fails to capture the variability in the true human distribution (e.g., true OOD subset). Right Table: Quantitative comparison showing intra- and inter-distance for clean and attacked data, indicating that the distance of human text is larger than the distance among LLM-generated text.

Existing LLM detection methods can be categorized into three groups: watermarking techniques [13], zero-shot statistical methods [14], and training-based approaches [15]. Watermarking methods embed identifiable signals into generated content during inference, offering strong detection when the model is cooperative, but failing entirely in black-box or adversarial settings. Zero-shot methods, which rely on distributional differences, are model-agnostic and do not require training, making them flexible and efficient. But their performance is often unstable across domains and text lengths. In contrast, training-based methods, typically implemented as supervised classifiers trained on labeled examples, often achieve higher detection accuracy. Nevertheless, all these methods that require labeled data face the challenge of poor generalization across text detection from different domains or unseen models.

Most existing detectors, whether zero-shot or supervised, treat the detection as a binary classification task, aiming to distinguish between the distributions of human-written and LLMs-generated content by identifying a threshold or optimal decision boundary between their distributions. However, this binary classification paradigm carries inherent limitations. While prior work [15] has demonstrated that machine-generated text often exhibits consistent statistical patterns suggesting that it can be reasonably modeled as a learnable distribution, a fundamental challenge lies in ***can human-written text truly be treated as a single, coherent distribution for the purpose of classification?*** Intuitively, human language is inherently diverse: each author, writing style, domain, or genre may follow distinct linguistic patterns, leading to a highly heterogeneous and dispersed distribution in the feature space. This diversity poses a critical problem for binary classification models: by treating human-written text as a single distribution, existing approaches risk oversimplifying the underlying variability. Such oversimplification can result in poor generalization, especially when encountering human texts from unseen domains or stylistic backgrounds.

Figure 1 Left figure illustrates the practical limitations of applying binary classification to this task. The in-distribution (ID) region, representing machine-generated text, is compact and well-covered by training data. However, human-written text exhibits variability as out-of-distribution samples, especially in the "true OOD" subset (shown as red triangles), which lies outside the classifier's detection boundary. As a result, a binary classifier (the red dashed line), trained on limited human samples, tends to overfit and produce a brittle decision boundary with poor generalization. Moreover, as shown in the Right Table in Figure 1, we compute the average cosine inter- and intra-distance of LLM-generated and human-written texts on the Deepfake dataset (Clean Data) and RAID dataset (Attacked Data). The embedding backbone we use is RoBERTa pretrained with contrastive loss. The results show that the Intra-distance of LLM is smaller than the human while the inter-distance between human and LLM is relatively large, supporting our OOD hypothesis that LLM text forms a coherent in-distribution cluster, while human-written texts, which span varied styles and domains, act as natural OOD examples.

In this work, we theoretically analyze the representation incompleteness of human text distribution and the corresponding failure reason of binary classifiers. Furthermore, to address the issue of

inherent asymmetry between human and machine-generated text distributions, we propose to recast machine-generated text detection as an out-of-distribution (OOD) detection task rather than a binary classification problem. OOD detection methods focus on modeling only the in-distribution region and treat any deviation from it as unknown samples. Under this new formulation, we treat machine-generated text as the in-distribution (ID) samples, assuming that it originates from a single or limited family of models with shared statistical characteristics. In contrast, human-written text is considered as out-of-distribution samples, reflecting its open-ended, diverse, and difficult-to-model nature. OOD detection approach yields a more principled and robust decision boundary that fits the open-ended diversity of human-authored texts (the blue dashed line in Figure 1 Left).

To validate our core hypothesis, that machine-generated texts form a learnable distributional core while human-written texts are distributional outliers, we systematically evaluate a range of established OOD detection methods. We implement this OOD detection framework by training a one-class or score-based classifier that models the distribution of machine-generated texts. We incorporate and validate different OOD detection methods including DeepSVDD [16], HRN [17] and Energy-based method [18]. At inference time, any input that significantly deviates from this learned distribution is flagged as OOD samples and considered to be human-written. Empirically, we compare our method against traditional supervised binary classifiers and recent zero-shot detectors. Remarkably, all OOD-based methods obtain superior detection performance. Specifically, the OOD-based method achieves 98.3% AUROC and AUPR with only 8.9% FPR95 on DeepFake dataset. Moreover, we test our method on *multilingual* and *attacked* text setting on M4 and RAID datasets, where our method achieves SoTA performance. Finally, we validate the generalizability of our method on unseen models and domain scenarios, highlighting the capacity to generalize beyond specific domains or stylistic boundaries. In summary, our contributions are as follows:

- We identify the fundamental topological and statistical distinctions between human-written and LLM-generated texts in the representation space and theoretically prove the representation incompleteness of human-written text distribution and the corresponding failure reason of binary classifiers.

- Based on the theoretical analysis, we propose reframing machine-generated text detection as an out-of-distribution (OOD) detection problem instead of a binary classification task. We solve this problem by designing a detection framework grounded in one-class and score-based learning that leverages the distributional convergence of LLM outputs.

- Empirically, we incorporate various OOD detection methods including DeepSVDD, HRN and Energy-based method. Extensive experiments on multiple benchmarks indicate that OOD-based methods achieve superior performance compared with baselines. Moreover, the strong detection performance on multilingual, attacked and unseen-model and -domain texts demonstrates the robustness and generalizability of our framework.

## 2 Related Works

**LLMs-generated Text Detection.** Existing approaches for detecting LLMs-generated text can be broadly categorized into three main categories [19]: watermarking [20, 21, 22, 13, 23], zero-shot methods [24, 25, 26, 27, 28, 29, 30, 31], and supervised training-based classifiers [32, 33, 34, 35, 36, 37, 38, 39]. Watermarking methods operate under a different paradigm compared to the other two by embedding identifiable patterns or signals into the text during model training, but is only feasible for model providers. In contrast, zero-shot and supervised training-based approaches are typical post hoc methods and both frame the detection task as a binary classification problem: given a candidate text, determine whether it was generated by an LLM (positive sample) or written by a human (negative sample). Zero-shot detectors, such as DetectGPT [14], DetectLLM [40], DNA-GPT [41], Fast-DetectGPT [25], Binoculars [42], BiScope [43], Raidar [44], and DALD [45], typically rely on the statistical analysis of the logits on distribution irregularities between human- and machine-written texts, and distinguish them using threshold-based decision rules. More recently, detection has also been approached through statistical hypothesis testing methods [46, 47] which reframe detection as a relative test problem. Supervised training-based methods generally achieve higher accuracy by training a binary classifier on labeled datasets containing human and machine-generated texts. These approaches often follow a pipeline that uses pretrained language models (e.g., RoBERTa, T5 ) as embedding extractors, followed by a classifier layer. Notable training based examples include GPT-Sentinel [48], GhostBuster [49], GPTZero [50]. Building on these backbones, RADAR [34]

adopts an adversarial training strategy, while CoCo [33] and DeTeCtive [15] leverage contrastive learning in the feature space to distinguish between clusters associated with different LLMs and human-authored text. Despite their effectiveness in specific detection scenarios, all of these methods fundamentally assume that human-written text, the negative class, can be modeled as a unified and coherent distribution. However, this assumption overlooks the inherent diversity and stylistic variability of human language, limiting generalization and robustness when applied to out-of-domain or heterogeneous human text samples.

**Out-of-Distribution Detection.** Out-of-distribution (OOD) detection [51] is a long-standing problem in machine learning, extensively studied in computer vision and anomaly detection [52, 53, 16, 54, 55, 56]. This task focuses on identifying inputs that deviate from the data distribution observed during training. A wide range of OOD detection techniques have been proposed. One-class learning approaches, such as One-Class SVM [57] and Support Vector Data Description (SVDD) [58], attempt to learn a tight boundary around ID samples without requiring any OOD data. In the deep learning era, variants such as DeepSVDD [16] and autoencoder-based models have extended these principles to high-dimensional representation spaces. Score-based methods [59, 60, 61, 62, 63, 64], including softmax score [65], and energy score [18], estimate confidence scores during inference to identify inputs with low model certainty. More recent trends include contrastive learning and representation clustering [66, 67], which aim to separate ID and OOD samples in embedding space without relying on labeled outliers. In the context of natural language processing (NLP), OOD detection presents unique challenges [68]. Pretrained language models such as BERT and RoBERTa have been used to extract embeddings, which are then analyzed using one-class or score-based techniques [69]. However, few studies have explored OOD detection in the context of LLM-generated text. Our work is the first to systematically apply OOD techniques to this detection problem, offering improved generalization and robustness across models.

## 3 Methods

In this section, we present the details of our proposed approach. Section 3.1 formalizes the problem of AI-generated text detection and motivates our reformulation of the task as an out-of-distribution (OOD) detection problem. Section 3.2 outlines the overall detection pipeline. Section 3.3 describes the specific OOD detection methods we adopt—DeepSVDD, HRN, and Energy-based method.

### 3.1 Problem Reformulation: Why Binary Classification Fails for Human Text Detection?

We aim to detect whether a given text is generated by a large language model (LLM) or authored by a human. Traditionally, this problem has been framed as a binary classification task, where the detector is trained to distinguish between human-written and machine-generated samples using supervised learning. Formally, let $\mathcal{X}$ be the text space, and consider a dataset $\mathcal{D} = \{(x, y) | x \in \mathcal{X}, y \in \{0, 1\}\}$ generated with class probabilities $q_M, q_H$ and in-class distributions $P_H, P_M$:

$$\mathbb{P}(y = 0) = q_M, \quad \mathbb{P}(y = 1) = q_H = 1 - q_M,$$
$$\mathbb{P}(x|y = 0) = P_M(x), \quad \mathbb{P}(x|y = 1) = P_H(x), \tag{1}$$

where the label $y = 0$ corresponds to machine-generated text, and $y = 1$ corresponds to human-generated text. The following concept characterizes the "correctness" of such a data distribution.

**Assumption 1** (Human-Machine Distinction Hypothesis)**.** We assume for any given text $x \in \mathcal{X}$, there exists an objective ground truth probability $\widehat{p}_M(x)$ that $x$ is machine-generated. We say a data distribution defined by the tuple $(q_M, q_H, P_M, P_H)$ is consistent with the ground truth $\widehat{p}_M(x)$ if, for $x \in \mathcal{X}$ and label $y \in \{0, 1\}$,

$$\frac{q_M \cdot P_M(x)}{q_H \cdot P_H(x)} = \frac{\widehat{p}_M(x)}{1 - \widehat{p}_M(x)}.$$

Note that this is perfectly in line with the Bayes rule: $\mathbb{P}(y = 0|x) = q_M P_M(x) / [q_M P_M(x) + q_H P_H(x)]$. The goal now is to train a classifier $f : \mathcal{X} \to \{0, 1\}$ that distinguishes between the two distributions:

$$f(x) \in \{0, 1\}, \quad \text{where} \quad \begin{cases} f(x) = 1 & \text{if } x \sim P_{\text{human}}, \\ f(x) = 0 & \text{if } x \sim P_{\text{machine}}. \end{cases} \tag{2}$$

In this setting, the classifier is typically trained with a cross-entropy loss, which encourages the model to maximize confidence over both classes. The training objective is:

$$\mathcal{L}_{\text{CE}}(f \mid \mathcal{D}) = -\mathbb{E}_{(x,\widehat{y}) \sim \mathcal{D}} \big[ \log p_\theta(y = \widehat{y} \mid x) \big] \tag{3}$$

$$= -q_M \mathbb{E}_{x \sim P_M} \big[ \log p_\theta(y = 0 \mid x) \big] - q_H \mathbb{E}_{x \sim P_H} \big[ \log p_\theta(y = 1 \mid x) \big]. \tag{4}$$

It can be proven (See Appendix) that the loss reaches minimum $\mathbb{E}_{(x,y) \sim \mathcal{D}}[\mathcal{H}(\widehat{p}_M(x))]$ when $p_\theta(y = 0 \mid x) = \widehat{p}_M(x)$, where $\mathcal{H}$ denotes entropy, thus encouraging the model to align itself with the ground truth label distribution. However, we posit that textual distributions generated by different human authors exhibit substantial heterogeneity, characterized by divergent stylistic patterns, domain-specific variations, and individualized linguistic fingerprints. To formalize this intuition, we consider an open-world human text distribution with intrinsic heterogeneity $\widehat{P}_H$, which is much more diverse than $P_H$, and state the following theorem.

**Theorem 2.** Consider training distribution $\mathcal{D} = (q_M, q_H, P_M, P_H)$ and real-world distribution $\widehat{\mathcal{D}} = (q_M, q_H, \widehat{P}_M, \widehat{P}_H)$ both consistent with the ground truth probability $\widehat{p}_M(\cdot)$. If the Pearson $\chi^2$ divergence $D := D_{\chi^2}(P_H || \widehat{P}_H)$ is large, then for an arbitrary suboptimality $\Delta$, there exists a binary classifier $f$, such that:

- The loss for $\mathcal{D}$ is close to optimal: $\mathcal{L}_{\text{CE}}(f \mid \mathcal{D}) < \mathbb{E}_{(x,y) \sim \mathcal{D}}[\mathcal{H}(\widehat{p}_M(x))] + \Delta$;

- The loss for $\widehat{\mathcal{D}}$ is high: $\mathcal{L}_{\text{CE}}(f \mid \widehat{\mathcal{D}}) > \mathbb{E}_{(x,y) \sim \widehat{\mathcal{D}}}[\mathcal{H}(\widehat{p}_M(x))] + 0.9 q_H (D + 1) \cdot \Delta$.

Detailed proof of this theorem and an analysis of the magnitude of $D$ can be found in the Appendix. Theorem 2 implies that the error rate of a binary classifier on the real-world human text distribution can be many times higher than the training error, due to the inherent heterogeneity of human-written text (e.g., diverse styles, domains, and author traits). Another way to pinpoint this problem is to consider a training dataset *almost consistent* with $\widehat{p}_M$, while showing high discrepancy at sparser text regions. A binary classifier trained on this dataset will inevitably overfit and fail to generalize to the real-world distribution. We present another theoretical result based on this line of thinking also in Appendix. Either way, this highlights a fundamental mismatch: binary classification assumes a closed-world setting, yet human-written text behaves as an open-world, high-variance distribution.

To address this, we propose to reframe the machine-generated text detection as an OOD detection problem, where the machine-generated distribution $P_M = P_{\text{in}}$ is modeled explicitly as a narrow, well-characterized distribution, and any deviation from it is treated as out-of-distribution $P_H = P_{\text{out}}$ without attempting to model it directly. This $P_{\text{out}}$ may cover texts from unseen domains, stylistic outliers, noisy generations, languages, authors, or other edge cases not represented in the training data. This perspective not only aligns better with the theoretical properties of text generation, but also provides a more robust framework for handling the diversity and unpredictability of human language.

### 3.2 Method Overview: OOD Detection Framework for Machine-Generated Text

To overcome the limitations of binary classification in the context of machine-generated text detection, we adopt an out-of-distribution (OOD) detection framework based on the one-class and score-based learning paradigm. Rather than learning to discriminate between two classes, our method focuses solely on modeling the in-distribution, i.e., machine-generated text, which is typically compact and consistent due to its origin from a specific LLM. Our framework consists of two main components: 1) a **Text Encoder** (e.g. RoBERTa) that maps each input text $x \in \mathcal{X}$ into a high-dimensional representation $z = \phi_\theta(x) \in \mathbb{R}^d$. 2) an **OOD Detector** that learns an OOD decision boundary or score by designed OOD loss. Figure 2 illustrates the overall pipeline by taking DeepSVDD as a primary example of the second stage. DeepSVDD learns a compact hypersphere around the in-distribution data in the embedding space. Overall, during the training stage, we compute a center point $c \in \mathbb{R}^d$ using only the embeddings of machine-generated samples, then optimize the encoder parameters using the following DeepSVDD loss:

$$\mathcal{L}_{\text{DSVDD}} = \frac{1}{N} \sum_{i=1}^{N} \| \phi_\theta(x_i) - c \|^2. \tag{5}$$

The objective is to compute and minimize the distance of machine-generated samples from the center, which is intuitively the radius of the hypersphere, thereby ensuring that embeddings of machine-generated texts lie close to the center. Unlike binary classification, which attempts to model both

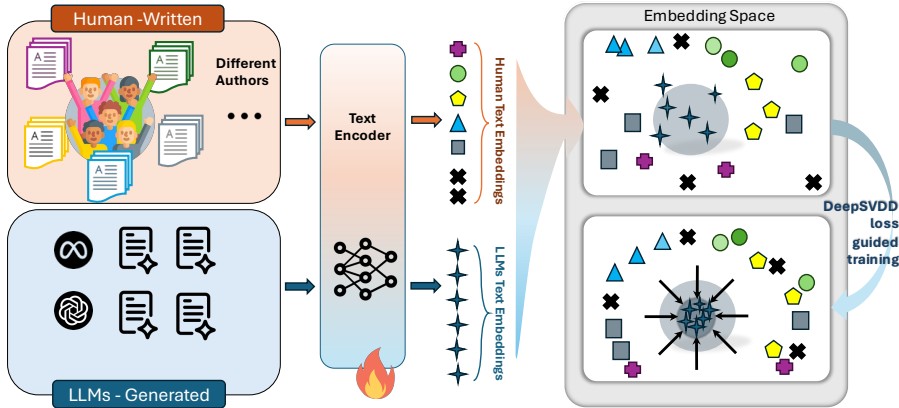

Figure 2: An overview of our proposed OOD detection pipeline under DeepSVDD method. This pipeline shows the process of training a text encoder with a DeepSVDD loss. (Right) Illustration of the learned embedding space: LLM-generated texts are enclosed within a hypersphere, while human-written texts fall outside.

classes under low-entropy supervision, our one-class method only learns the compact support of machine text, while allowing high-entropy human texts to remain unconstrained, treating any outlier as likely authored by a human.

Besides, to further enhance the model's ability to distinguish between human- and machine-generated text, we incorporate a contrastive loss term based on SimCLR[70] during training, as shown in Equation 6. $q$ is the current sample. $Z^+$ is the collection of positive samples, while $Z^-$ is the collection of negative samples. $N_{Z^+}$ is the size of positive sample collection. $\tau$ refers to the temperature coefficient. $S(\cdot)$ is the similarity computation, which is cosine similarity in our experiments.

$$\mathcal{L}_{\text{contrastive}} = -\log \frac{\exp\left(\sum_{z \in Z^+} \frac{S(q,z)}{\tau} / N_{Z^+}\right)}{\exp\left(\sum_{z \in Z^+} \frac{S(q,z)}{\tau} / N_{Z^+}\right) + \sum_{z \in Z^-} \exp\left(\frac{S(q,z)}{\tau}\right)}. \tag{6}$$

### 3.3 Alternative OOD Detection Losses: HRN and Energy-Based Methods

While we describe DeepSVDD as a representative OOD detection approach, our framework is model-agnostic and supports a variety of one-class and score-based OOD detection methods. In this section, we present two additional techniques HRN [17] and Energe-based OOD detection [18] that instantiate the same pipeline with alternative OOD loss formulations. These methods are designed to improve detection reliability under varying conditions. Each is seamlessly integrated into the same architecture described in Section 3.2, replacing the DeepSVDD loss with their OOD loss function.

**HRN** is a one-class learning method designed to improve robustness and generalization in OOD detection. HRN only has one class/head in the output, it uses Sigmoid($f(\phi(x))$) as a score or the probability of given text $x$ belonging to the in-class. During training, learning the given class in the distribution with its training data, its one-class loss is:

$$\mathcal{L}_{\text{HRN}} = \mathbb{E}_{x \sim \mathcal{D}_{\text{in}}^{\text{train}}} \left[-\log(\text{Sigmoid}(f(\phi(x))))\right] + \lambda \cdot \|\nabla_x f(\phi(x))\|_2^n, \tag{7}$$

where $f$ is a one-class classifier, $\phi(x)$ is the high-dimensional embedding, Sigmoid$(\cdot) \in (0, 1)$ is a score mapping the one class head to logit, and $n$ and $\lambda$ are hyper-parameters controlling the strength of the penalty and balancing the regularization respectively. By adding an H-Regularization term with 2-Norm instance-level normalization to the Negative Log Likelihood for one class, this HRN loss improves over prior approaches by penalizing excessive sensitivity to specific input directions, promoting more robust and generalizable decision functions.

**Energy-based** OOD detection methods assign an energy score $E(x)$ to each input, indicating its compatibility with the in-distribution. Lower energy implies a higher likelihood of being in-

distribution. The energy of a sample is defined from softmax logits $f(x)$ as:

$$E(x) = -\log \sum_i \exp(f_i(x)), \tag{8}$$

where $f_i(\cdot)$ indicates the model logits for class $i$. We adopt an energy-based learning objective that explicitly encourages an energy gap during training. The learning objective is:

$$\mathcal{L}_{\text{energy}} = \mathbb{E}_{(x,y)\sim \mathcal{D}_{\text{in}}^{\text{train}}} \left[ -\log F_y(x) \right]$$
$$+ \lambda \cdot \left\{ \mathbb{E}_{x\sim \mathcal{D}_{\text{in}}^{\text{train}}} \left[ \max(0, E(x) - m_{\text{in}})^2 \right] + \mathbb{E}_{x\sim \mathcal{D}_{\text{out}}^{\text{train}}} \left[ \max(0, m_{\text{out}} - E(x))^2 \right] \right\}, \tag{9}$$

where $F_y(x)$ is the softmax output of the classifier. Importantly, the softmax classifier is trained only on in-distribution data—i.e., machine-generated texts from multiple sources—using cross-entropy loss to distinguish between different LLM generators. The second is an energy regularization term, that combines two squared hinge loss terms with separate margin hyperparameters $m_{\text{in}}$ and $m_{\text{out}}$. This loss penalizes in-distribution samples whose energy exceeds a predefined margin $m_{\text{in}}$, and simultaneously penalizes OOD samples with energy lower than another threshold $m_{\text{out}}$. The resulting model learns a contrastively shaped energy surface, creating a clearer margin between ID and OOD samples in energy space.

### 3.4 Overall Training Objectives

The final training objective thus combines the OOD loss with a supervised contrastive objective:

$$\mathcal{L}_{\text{Total}} = \alpha \cdot \mathcal{L}_{\text{OOD}} + \beta \cdot \mathcal{L}_{\text{contrastive}}, \quad \mathcal{L}_{\text{OOD}} \in \{\mathcal{L}_{\text{DSVDD}}, \mathcal{L}_{\text{HRN}}, \mathcal{L}_{\text{energy}}\} \tag{10}$$

where $\alpha$ and $\beta$ is a weighting coefficient. This joint training scheme allows the encoder to benefit from both unsupervised in-distribution modeling and supervised human–machine discrimination, leading to more robust representation learning.

**Inference.** For Deep-SVDD, we compute an OOD score for a given text using the distance from the center, if the score exceeds a threshold, the text is flagged as out-of-distribution and thus considered human-written. As for HRN, during inference time, texts with low sigmoid scores are considered as OOD samples, which is human-written text. Finally, for energy-based method, the energy score $E(x)$ is computed for a given input using the log-sum-exp of the softmax logits. An input is classified as out-of-distribution if its negative energy score $-E(x)$ falls below a predefined threshold.

## 4 Experiments

### 4.1 Experimental Setup

**Datasets.** We test our method on three widely-used and challenging datasets including DeepFake [71], M4 [72], and RAID [73]. The Deepfake dataset comprises text generated by 27 large language models (LLMs) alongside human-written content sourced from multiple websites across 10 distinct domains, totaling 332K training and 57K test samples. It defines six diverse evaluation scenarios, including cross-domain, unseen domain, and model detection settings. Moreover, the M4 dataset is a comprehensive benchmark spanning multiple domains, models, and languages, comprising data from 8 large language models (LLMs), 6 domains, and 9 languages. We test our method and other baselines on multilingual settings, which include 157K training and 42K testing data. The test dataset of M4 also includes the unseen models from the training set. Finally, RAID is a large-scale LLM detection benchmark with normal AI-generated texts and different attacked texts including paraphrase, synonym swap, misspelling, and so on. We hold out 10% of training data as validation set for evaluation. During training, we filter the attacked samples in the training set.

**Evaluation Metrics.** To comprehensively evaluate the detection results, we follow DNA-GPT [41], FastDetectGPT [25] and DALD [45] and report the accuracy in the area under the receiver operating characteristic (AUROC) and the area under the precision and recall (AUPR) to evaluate the performance of all methods. Moreover, we follow [18] to measure the false positive rate (FPR95) when true positive rate is at 95%.

**Baselines.** We compare our method with both zero-shot and training-based methods. For zero-shot methods, we compare with DetectGPT [14], FastDetectGPT [25], DNA-GPT [41], Binoculars [42],

Table 1: Detection result comparison of our method and baselines. Our method outperforms all previous zero-shot and training-based methods and achieves SoTA performance. †: Due to the low detection speed, we randomly select 10K samples from each test set for evaluation.

| Method | DeepFake | | | M4-multlingual | | | RAID | | |
|---|---|---|---|---|---|---|---|---|---|
| | AUROC ↑ | AUPR↑ | FPR95↓ | AUROC↑ | AUPR↑ | FPR95↓ | AUROC↑ | AUPR↑ | FPR95↓ |
| DetectLLM† [40] | 59.4 | 62.5 | 95.0 | 86.7 | 83.2 | 46.0 | 77.6 | 80.5 | 78.6 |
| DetectGPT† [14] | 58.1 | 59.4 | 90.2 | 70.5 | 67.3 | 80.4 | 59.5 | 77.3 | 70.5 |
| DNA-GPT† [41] | 57.9 | 56.0 | 96.5 | 66.2 | 61.2 | 74.1 | 61.3 | 97.3 | 95.8 |
| F-DetectGPT[25] | 60.3 | 70.8 | 99.9 | 87.7 | 91.6 | 72.8 | 77.6 | 99.2 | 99.3 |
| Binoculars [42] | 61.7 | 52.8 | 66.7 | 87.7 | 76.6 | 27.8 | 82.4 | 7.3 | 40.4 |
| DALD [45] | 56.2 | 55.0 | 97.4 | 63.2 | 66.6 | 64.6 | 59.4 | 67.1 | 64.6 |
| Glimpse [74] | 69.5 | 74.5 | 91.4 | 88.2 | 89.0 | 52.8 | 79.5 | 99.1 | 81.2 |
| GPTZero [50] | 59.3 | 70.4 | 92.4 | 67.5 | 76.7 | 89.8 | 62.7 | 68.5 | 92.0 |
| RADAR [34] | 61.2 | 55.3 | 86.3 | 68.3 | 76.1 | 92.1 | 82.9 | 69.2 | 64.1 |
| GhostBuster [49] | 52.9 | 51.1 | 93.2 | 62.5 | 63.2 | 85.6 | 66.7 | 98.1 | 82.4 |
| BiScope [43] | 79.2 | 85.0 | 84.3 | 83.0 | 90.8 | 84.8 | 78.0 | 71.1 | 86.8 |
| DeTeCtive [40] | 95.6 | 97.1 | 16.1 | 93.1 | 95.3 | 52.4 | 85.1 | 55.7 | 74.3 |
| Ours (D-SVDD) | **98.3** | **98.3** | 8.9 | **96.4** | 96.3 | 18.2 | **94.7** | 73.0 | 38.3 |
| Ours (HRN) | 97.0 | 97.2 | **4.4** | **96.4** | **96.7** | 24.9 | 88.1 | 70.1 | 57.5 |
| Ours (Energy) | 97.6 | 97.2 | 12.3 | 96.2 | 95.8 | **10.7** | 93.6 | **99.7** | **28.3** |

DetectLLM LRR and NPR [40], DALD [45], BiScope [43], and Glimpse [74]. For training-based methods, we include the comparison with RADAR [34], GhostBuster [49], GPT-Zero, and DeTeCtive [15]. We utilize the official implementation of each baseline. More information about the baselines can be found in the supplementary material.

**Implementation Details.** We adopt three OOD detection methods including DeepSVDD, HRN and Energy-based methods for our experiments. For DeepSVDD, we use the machine texts from the training set to compute the initial center and compute the corresponding loss with the center. For HRN, follow its original setting by training a one-class classifier per model family using its corresponding data as the positive class and averaging their scores at inference time. For energy-based method, we follow [18] to choose hyper-parameters, where $m_{in} = -27$ and $m_{out} = -2$. DeepSVDD is trained from scratch, and HRN and Energy load the pre-trained weights of DeTeCtive. The learning-rate is set as 2e-5 and the optimizer is Adam [75] with $\beta_1 = 0.9$ and $\beta_2 = 0.98$. The loss weights $\alpha$ and $\beta$ are set as 1 in our experiments. We train all model for 20 epochs. All experiments are conducted on 2*A100 GPUs.

## 4.2 Main Results

We conduct a comprehensive evaluation of our methods against a diverse set of baselines on two challenging benchmarks: DeepFake and M4-multilingual, as present in Table 1. On the DeepFake dataset, our DeepSVDD model achieves the highest AUROC (98.3) and AUPR (98.3), surpassing all other training-based methods including GPT-Zero, RADAR, GhostBuster. Moreover, our method substantially outperforms the strongest training-based baseline, DeTeCtive, which scores 95.6 AUROC and 97.1 AUPR. Additionally, DeepSVDD reduces the FPR95 from 16.1% to 8.9%, indicating a much lower false positive rate at high sensitivity. Our HRN and Energy-based variants also perform competitively, with HRN attaining the lowest FPR95 (4.4%) among all methods while maintaining high precision (97.2 AUPR). In the M4-multilingual benchmark, where detectors must generalize across multiple languages and domains, our methods again lead in all metrics. DeepSVDD and HRN both reach an AUROC of 96.4, surpassing DeTeCtive's 93.1. HRN also delivers the best AUPR (96.7), while the Energy-based method achieves a strong balance between recall and reliability, with a notably low FPR95 of 10.7, compared to DeTeCtive's much higher 52.4. These results demonstrate the effectiveness and robustness of our approaches in *cross-lingual* and *domain-shifted* settings.

We also compare against zero-shot methods including FastDetectGPT and Binoculars, two recent approaches designed to detect LLM-generated content without task-specific training. While FastDetectGPT offers fast and training-free deployment, it suffers from significant performance limitations. On DeepFake, it achieves only 78.3 AUROC and 60.8 FPR95—over 20 points worse in AUROC and nearly seven times higher in FPR95 than DeepSVDD. Binoculars, a more recent zero-shot method, performs somewhat better with 88.1 AUROC and 41.2 FPR95, but still underperforms our models by a large margin. On M4-multilingual, FastDetectGPT's AUROC drops to 66.1 and FPR95 increases to 74.7, while Binoculars shows moderate improvement (79.6 AUROC, 64.9 FPR95), yet both remain significantly below the performance of our proposed detectors.

Table 2: Detection results in comparison of baseline and our method in unseen domain and model settings. With different OOD heads, our method surpasses the DeTeCtive in all settings, demonstrating the generalizability of our method.

| Method | Unseen Domain | | | Unseen Model | | |
|---|---|---|---|---|---|---|
| | AUROC ↑ | AUPR↑ | FPR95↓ | AUROC↑ | AUPR↑ | FPR95↓ |
| DeTeCtive | 76.5 | 87.5 | 89.2 | 84.3 | 88.1 | 70.8 |
| Ours (D-SVDD) | 97.9 | **97.8** | 10.2 | 93.4 | 92.3 | **25.8** |
| Ours (HRN) | **98.0** | **97.8** | 12.0 | **95.2** | **96.3** | 30.9 |
| Ours (Energy) | 96.0 | 93.2 | **8.3** | 90.2 | 91.3 | 38.1 |

Table 3: Ablation study. The binary classification head is compared with each of our components.

| Method | DeepFake | | | M4-multlingual | | | RAID | | |
|---|---|---|---|---|---|---|---|---|---|
| | AUROC ↑ | AUPR↑ | FPR95↓ | AUROC↑ | AUPR↑ | FPR95↓ | AUROC↑ | AUPR↑ | FPR95↓ |
| Classification Head | 89.0 | 94.0 | 75.8 | 84.6 | 91.2 | 83.2 | 84.5 | 71.9 | 83.5 |
| D-SVDD Loss | **98.3** | **98.3** | 8.9 | **96.4** | 96.3 | 18.2 | **94.7** | 73.0 | 38.3 |
| HRN Loss | 97.0 | 97.2 | **4.4** | **96.4** | **96.7** | 24.9 | 88.1 | 65.9 | 70.0 |
| Energy Loss | 97.6 | 97.2 | 12.3 | 96.2 | 95.8 | **10.7** | 93.6 | **99.7** | **28.3** |

Overall, our methods consistently outperform both training-based and zero-shot baselines across all metrics and benchmarks. The substantial gains in AUROC and AUPR, along with dramatic reductions in FPR95, demonstrate that our approach offers a robust and generalizable solution for detecting LLM-generated text, even in multilingual and domain-shifted environments.

## 4.3 Experimental Analysis

**Attack Robustness.** We also provide the results on RAID dataset, as shown in Table 1. RAID is specifically designed to evaluate the robustness of LLM-generated text detectors under adversarial perturbations including paraphrasing, synonym substitution, sentence reordering, and so on. Our proposed methods exhibit strong robustness to adversarial attacks, significantly outperforming both training-based and zero-shot baselines. Notably, the energy-based model achieves the highest AUPR (99.7), indicating exceptional precision-recall performance even under strong perturbations. This suggests that the model remains confident and discriminative even when the surface form of the generated text is substantially altered. Additionally, the DeepSVDD variant attains the highest AUROC (94.7) among all methods, demonstrating that it maintains strong overall ranking performance in the presence of adversarial noise. In contrast, the baseline methods show siginificant degradation on RAID. For example, DeTeCtive, while effective on clean data, sees its AUROC drop to 89.3 with a much higher FPR95 (58.9), indicating frequent misclassifications of perturbed generated samples as human-written. The zero-shot method FastDetectGPT performs particularly poorly, with only 59.2 AUROC and a very high FPR95 of 75.1, showing that it lacks the structural resilience needed for attack scenarios. Even Binocular, which performs better than FastDetectGPT, achieves only 72.4 AUROC and 64.3 FPR95—well below those of our proposed models. These results highlight a key strength of our approach: robust generalization to adversarially manipulated text. The combination of representation learning (in DeepSVDD and HRN) and energy-based scoring mechanisms provides a defense against common attack strategies that exploit shallow decision boundaries or overfitting.

**Generalizability.** To show the generalizability of our method, we test our method on DeepFake dataset in *Unseen Domains* and *Unseen Models* settings, which is split from [15]. The results are shown in Table 2. In the unseen domain scenario, all variants of our method significantly outperform DeTeCtive. Specifically, our HRN-based method achieves the highest AUROC (98.0) and a near-optimal AUPR (97.8), with a considerably lower FPR95 (12.0) compared to DeTeCtive's 89.2. The DeepSVDD variant also performs competitively, achieving an AUROC of 97.9 and the lowest FPR95 of 10.2. Notably, the energy-based approach achieves the best FPR95 (8.3), suggesting that it is especially effective at reducing false positives, even though its AUROC (96.0) is slightly lower than the others. In the unseen model setting, similar performance is observed. The strong performance underscores our method's robustness in identifying the instances from unseen models and domains with minimal false positives and demonstrates the generalizability of our method.

**Practical Insights on Method Choice.** According to the empirical results above, we can draw a conclusion and provide the insights about how to choose OOD detection methods: 1) In scenario which requires attack robustness, energy-based methods yield the best performance; 2) In scenario

which requires better generalizability like unseen model and domains, HRN would be a better choice; 3) DeepSVDD balances both scenario.

**Ablation Study.** We conduct an ablation study of our method on DeepFake, M4, and RAID datasets. The results are presented in Table 3. We compare our loss functions including D-SVDD, HRN and Energy against a standard binary classification head across three datasets. Across all benchmarks, our methods significantly outperform the classification head, which suffers from high FPR95 values (75.8–83.5). Notably, HRN achieves the lowest FPR95 (4.4) on DeepFake, while Energy loss attains the best AUPR (99.7) and lowest FPR95 (28.3) on RAID. All variants also consistently improve AUROC and AUPR, confirming the effectiveness of our loss designs in enhancing LLM detection performance and reducing false positives.

## 5 Conclusion

In this paper, we theoretically analyze the failure case and reason for treating LLM detection tasks as binary classification tasks and propose to transform the task to an out-of-distribution detection task. Moreover, we propose a novel LLM detection framework based on the OOD detection loss. We validate the effectiveness of our proposed framework with DeepSVDD, HRN and energy-based method, achieving SoTA performance on multiple benchmarks including multilingual, attacked and unseen-model and -domain scenarios.

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

# Contents

# A Proof of Theoretical Results

In this section, we provide formal justification for Theorem 2, discuss the magnitude of Pearson $\chi^2$ divergence, and provide an alternative theorem to further characterize the inevitability of overfitting binary classifiers to biased human-written text distributions.

## A.1 Preliminaries

We use integral $\int_{x \in \mathcal{X}} \cdot \, dx$ to equivalently denote integral or summation over the data space $\mathcal{X}$. For the ground truth classifier $\widehat{p}_M(x)$, we introduce the following auxiliary definitions to assist the writing:

$$\widehat{p}_H(x) := 1 - \widehat{p}_M(x)$$
$$\widehat{p}(y = 1|x) := \widehat{p}_H(x), \qquad \widehat{p}(y = 0|x) := \widehat{p}_M(x).$$

Towards a proof of the theorem, we first analyze the cross-entropy loss $\mathcal{L}_{\mathrm{CE}}$:

$$\mathcal{L}_{\mathrm{CE}}\big(f_\theta | \mathcal{D}\big) = -q_M \mathbb{E}_{x \sim P_M}\big[\log p_\theta(y = 0 \mid x)\big] - q_H \mathbb{E}_{x \sim P_H}\big[\log p_\theta(y = 1 \mid x)\big]$$

$$= -\int_{x \in \mathcal{X}}\bigg[q_M P_M(x) \log p_\theta(y = 0 \mid x) + q_H P_H(x) \log p_\theta(y = 1 \mid x)\bigg] dx$$

$$= -\int_{x \in \mathcal{X}}\bigg[\widehat{p}_M(x) \log p_\theta(y = 0 \mid x) + \big(1 - \widehat{p}_M(x)\big) \log p_\theta(y = 1 \mid x)\bigg] P_{\mathcal{D}}(x) \, dx$$

$$= \int_{x \in \mathcal{X}} \mathcal{H}\big(\widehat{p}(\cdot|x), p_\theta(\cdot|x)\big) P_{\mathcal{D}}(x) \, dx, \tag{11}$$

where the third equality comes from Assumption 1, $P_{\mathcal{D}}(x) := q_M P_M(x) + q_H P_H(x)$ denotes the joint probability at $x$ of dataset $\mathcal{D}$. and $\mathcal{H}(\cdot, \cdot)$ denotes the cross-entropy between two distributions. From the properties of cross-entropy, this is minimized when $\widehat{p} = p_\theta$, i.e. when the model predictions exactly match the ground truth, with minimal value

$$\min_\theta \mathcal{L}_{\mathrm{CE}}\big(f_\theta | \mathcal{D}\big) = \mathbb{E}_{x \sim \mathcal{D}} \mathcal{H}\big(\widehat{p}(\cdot|x)\big). \tag{12}$$

We also formally define The Pearson $\chi^2$ divergence below for future reference.

**Definition 3.** For two distributions $P_1, P_2$ over set $\mathcal{X}$, the Pearson $\chi^2$ divergence between $P_1$ and $P_2$ is defined as

$$D_{\chi^2}\big(P_1 \| P_2\big) = \int_{x \in \mathcal{X}} \frac{\big(P_1(x) - P_2(x)\big)^2}{P_2(x)} \, dx = \int_{x \in \mathcal{X}} \frac{P_1^2(x)}{P_2(x)} \, dx - 1 \tag{13}$$

## A.2 Proof of Theorem 2

The basic idea behind this proof is to find a classifier $f_\theta$ that properly accentuates the difference between data distributions $\mathcal{D}$ and $\widehat{\mathcal{D}}$, ensuring bounded suboptimality on the training set $\mathcal{D}$ while maximizing suboptimality on the real-world set $\widehat{\mathcal{D}}$. This is achieved through relatively enlarging the errors in regions where $\mathcal{D}$ has lower density compared to $\widehat{\mathcal{D}}$ and vice versa, hence blowing up the loss measured under $\widehat{\mathcal{D}}$ compared to $\mathcal{D}$.

*Proof of Theorem 2.* Let $\Delta(x) = \mathcal{H}\big(\widehat{p}(\cdot|x), p_\theta(\cdot|x)\big) - \mathcal{H}\big(\widehat{p}(\cdot|x)\big)$ be the suboptimality of $f_\theta$ at $x$. From (11),

$$\mathcal{L}_{\mathrm{CE}}\big(f_\theta | \mathcal{D}\big) - \mathbb{E}_{x \sim \mathcal{D}} \mathcal{H}\big(\widehat{p}(\cdot|x)\big) = \int_{x \in \mathcal{X}} \Delta(x) P_{\mathcal{D}}(x) \, dx, \tag{14}$$

$$\mathcal{L}_{\mathrm{CE}}\big(f_\theta | \widehat{\mathcal{D}}\big) - \mathbb{E}_{x \sim \widehat{\mathcal{D}}} \mathcal{H}\big(\widehat{p}(\cdot|x)\big) = \int_{x \in \mathcal{X}} \Delta(x) P_{\widehat{\mathcal{D}}}(x) \, dx, \tag{15}$$

Since cross-entropy is unbounded for $p_\theta(\cdot|x)$ given any $\widehat{p}(\cdot|x)$, we can choose a classifier $f_\theta$ such that $\Delta(x) = \Delta_0 \times P_{\widehat{\mathcal{D}}}(x)/P_{\mathcal{D}}(x)$ for any $x \in \mathcal{X}$ given a target suboptimality $\Delta_0$. With this we have the training suboptimality

$$\int_{x \in \mathcal{X}} \Delta(x) P_{\mathcal{D}}(x) \, dx = \int_{x \in \mathcal{X}} \Delta_0 P_{\widehat{\mathcal{D}}}(x) \, dx = \Delta_0,$$

while the loss on ground truth dataset $\widehat{\mathcal{D}}$ is

$$\int_{x \in \mathcal{X}} \Delta(x) P_{\widehat{\mathcal{D}}}(x)\,dx = \int_{x \in \mathcal{X}} \Delta_0 \times \frac{P_{\widehat{\mathcal{D}}}^2(x)}{P_{\mathcal{D}}(x)}\,dx$$
$$= \Delta_0 \int_{x \in \mathcal{X}} \frac{\left(q_M \widehat{P}_M(x) + q_H \widehat{P}_H(x)\right)^2}{q_M P_M(x) + q_H P_H(x)}\,dx. \tag{16}$$

Now notice that from Assumption 1, the ratios

$$\frac{q_M \widehat{P}_M(x)}{q_H \widehat{P}_H(x)} = \frac{q_M P_M(x)}{q_H P_H(x)} = \frac{\widehat{p}_M(x)}{\widehat{p}_H(x)},$$

therefore we can write (16) as

$$\int_{x \in \mathcal{X}} \Delta(x) P_{\widehat{\mathcal{D}}}(x)\,dx = \Delta_0 \int_{x \in \mathcal{X}} \left(\frac{\widehat{p}_M(x)}{\widehat{p}_H(x)} + 1\right) \cdot q_H \frac{\widehat{P}_H^2(x)}{P_H(x)}\,dx$$
$$\geq \Delta_0 \int_{x \in \mathcal{X}} q_H \frac{\widehat{P}_H^2(x)}{P_H(x)}\,dx$$
$$= q_H \Delta_0 \left[D_{\chi^2}\left(\widehat{P}_H \| P_H\right) + 1\right],$$

thus finishing the proof. $\qquad\square$

## A.3   Analysis of the Pearson $\chi^2$ Divergence

The effectiveness of Theorem 2 depends on the magnitude of the Pearson $\chi^2$ divergence $D_{\chi^2}\left(\widehat{P}_H \| P_H\right)$: the larger this divergence, the larger the cross-entropy loss for a linear classifier well-fitted to the training set. Here we relate this value to the intuition that the training human-text data distribution is biased and cannot reflect open-world distribution.

We model the intuition of "open-world" distribution $\widehat{P}_H$ by considering a training distribution $P_H$ that is a **shifted biased distribution** of $\widehat{P}_H$. More concretely, this means that there is a subset $\mathcal{X}_0$ of $\mathcal{X}$, such that

$$P_H(x) = \begin{cases} C_1 \widehat{P}_H(x), x \in \mathcal{X}_0, \\ C_2 \widehat{P}_H(x), x \notin \mathcal{X}_0, \end{cases}$$

where $C_1, C_2$ are constants such that $C_1 \mu_{\widehat{P}_H}(\mathcal{X}_0) + C_2 \left[1 - \mu_{\widehat{P}_H}(\mathcal{X}_0)\right] = 1$ and $C_1 \gg C_2$, and $\mu_{\widehat{P}_H}$ denotes the measure with respect to $\widehat{P}_H$. Such a distribution follows the same probability ratios as $\widehat{P}_H$ within and without $\mathcal{X}_0$ respectively, but is very close to 0 outside $\mathcal{X}_0$, hence most of the samples come from the partial region $\mathcal{X}_0$. This is of course only an approximation of a realistic scenario, but it captures the essence of training data being only a partial observation of the whole picture. With this, the Pearson $\chi^2$ divergence is

$$D_{\chi^2}\left(\widehat{P}_H \| P_H\right) = \int_{x \in \mathcal{X}} \frac{\widehat{P}_H^2(x)}{P_H(x)}\,dx - 1$$
$$= \int_{x \in \mathcal{X}_0} \frac{1}{C_1} \widehat{P}_H(x)\,dx + \int_{x \in \mathcal{X} \setminus \mathcal{X}_0} \frac{1}{C_2} \widehat{P}_H(x)\,dx - 1$$
$$= \frac{\mu_{\widehat{P}_H}(\mathcal{X}_0)}{C_1} + \frac{1 - \mu_{\widehat{P}_H}(\mathcal{X}_0)}{C_2} - 1,$$

which **blows up to infinity** inverse-linearly as $C_2 \to 0$, corresponding to the likely case where the dataset is heavily biased towards a partial region of the text data space.

## A.4   Alternative View: Binary Classifiers Overfit to Slightly Defected Datasets

While theoretically reasonable, Assumption 1 usually does not hold for many data distributions. For example, if a training human-text distribution consists only of one "style" of texts, then the probability

that some human-text $x^*$ of another style is sampled from this distribution would be close to $0$, even smaller than the probability that $x^*$ is sampled from the machine process, i.e. $P_H(x^*) \ll P_M(x^*)$. This leads to a violation of Assumption 1 in that the Bayes probability that $x^*$ is human-generated does not match $\widehat{p}_M(x)$. In this scenario, binary classifiers actually can overfit to the training dataset and fail to generalize by a nonzero gap.

To illustrate this, we characterize this dataset defect in the following definition in place of Assumption 1.

**Definition 4.** For a machine-human text data distribution $\mathcal{D} = \{q_M, q_H, P_M, P_H\}$, define its **posterior binary classifier** to be

$$p_\mathcal{D}(y = 1|x) = \frac{q_M P_M(x)}{q_M P_M(x) + q_H P_H(x)},$$

$$p_\mathcal{D}(y = 0|x) = \frac{q_H P_H(x)}{q_M P_M(x) + q_H P_H(x)}.$$

We define the **kwality** of $\mathcal{D}$ as

$$\mathbf{K}(\mathcal{D}) := \mathbb{E}_{x \sim P_\mathcal{D}(x)} D_{\mathrm{KL}}\big[\widehat{p}(\cdot|x)\big\|p_\mathcal{D}(\cdot|x)\big],$$

which measures the expected KL-divergence between $p_\mathcal{D}$ and $\widehat{p}$ over joint text distribution $P_\mathcal{D}(x) = q_M P_M(x) + q_H P_H(x)$, and say dataset $\mathcal{D}$ is $\delta$-**suboptimal** if its kwality $\mathbf{K}(\mathcal{D}) < \delta$.

**Remark 5.** A few things are worthy of note:

- When the posterior distribution $p_\mathcal{D} \to \widehat{p}$, kwality reaches its minimum value $0$, while higher kwality values correspond to deviations in labeling and hence degradation of dataset.

- The idea behind this definition is that, realistically we cannot expect the training dataset to be perfectly accurate, and kwality more or less measures the deviation of the model from the ground truth classifier.

- The expectation in kwality is taken over text distribution $P(x)$, which means mislabeling data in *sparser* text regions are less detrimental to kwality than *denser* regions. This aligns with the fact that sparse regions contain less data points within the sampled dataset, where accuracy cannot be guaranteed, but also warrants less attention from the learning model.

Our main result is that fully training models on a near-optimal dataset can lead to overfitting, and result in a large validation loss on open-world data distribution.

**Theorem 6.** There exists a $\delta$-suboptimal dataset $\mathcal{D}$ such that for any binary classifier that achieves optimal cross-entropy loss on $\mathcal{D}$, the generalization loss on open-world dataset $\widehat{\mathcal{D}}$ which complies with $\widehat{p}$ (Assumption 1) is at least $\delta D_{\chi^2}\big(P_\mathcal{D}\big\|P_{\widehat{\mathcal{D}}}\big)$.

*Proof.* From optimal cross-entropy loss, we get from (11) that

$$0 = \mathcal{L}_{\mathrm{CE}}\big(f_\theta|\mathcal{D}\big) - \mathbb{E}_{x \sim \mathcal{D}}\mathcal{H}\big(p_\mathcal{D}(\cdot|x)\big)$$

$$= \int_{x \in \mathcal{X}} \left[\mathcal{H}\big(p_\mathcal{D}(\cdot|x), p_\theta(\cdot|x)\big) - \mathcal{H}\big(p_\mathcal{D}(\cdot|x)\big)\right] P_\mathcal{D}(x)\, dx,$$

which means for any $x \in \mathcal{X}$,

$$\mathcal{H}\big(p_\mathcal{D}(\cdot|x), p_\theta(\cdot|x)\big) = \mathcal{H}\big(p_\mathcal{D}(\cdot|x)\big) \Leftrightarrow p_\theta(\cdot|x) = p_\mathcal{D}(\cdot|x).$$

Now consider the generalization loss:

$$\mathcal{L}_{\mathrm{CE}}\big(f_\theta|\widehat{\mathcal{D}}\big) - \mathbb{E}_{x \sim \widehat{\mathcal{D}}}\mathcal{H}\big(\widehat{p}(\cdot|x)\big)$$

$$= \int_{x \in \mathcal{X}} \left[\mathcal{H}\big(\widehat{p}(\cdot|x), p_\theta(\cdot|x)\big) - \mathcal{H}\big(\widehat{p}(\cdot|x)\big)\right] P_{\widehat{\mathcal{D}}}(x)\, dx$$

$$= \int_{x \in \mathcal{X}} D_{\mathrm{KL}}\big[\widehat{p}(\cdot|x)\big\|p_\mathcal{D}(\cdot|x)\big] P_{\widehat{\mathcal{D}}}(x)\, dx,$$

Since KL-divergence is unbounded, we can choose $\widehat{p}(\cdot|x)$ for each $x \in \mathcal{X}$ such that $D_{\mathrm{KL}}\big[\widehat{p}(\cdot|x)\big\|p_{\mathcal{D}}(\cdot|x)\big] = \delta P_{\widehat{\mathcal{D}}}(x)/P_{\mathcal{D}}(x)$, which guarantees $\mathbf{K}(\mathcal{D}) = \delta$, while

$$\mathcal{L}_{\mathrm{CE}}\big(f_\theta|\widehat{\mathcal{D}}\big) - \mathbb{E}_{x\sim\widehat{\mathcal{D}}}\mathcal{H}\big(\widehat{p}(\cdot|x)\big) \geq \int_{x\in\mathcal{X}} \delta \times \frac{P_{\widehat{\mathcal{D}}}^2(x)}{P_{\mathcal{D}}(x)}\,dx = \delta D_{\chi^2}\big(P_{\mathcal{D}}\big\|P_{\widehat{\mathcal{D}}}\big).$$

$\square$

# B   Detail of Experimental Setup

## B.1   Datasets

In this section, we discuss more details of our dataset in our experiments.

- **DeepFake**. The Deepfake dataset comprises text generated by 27 large language models (LLMs) alongside human-written content sourced from multiple websites across 10 distinct domains, totaling 332K training and 57K test samples. It defines six diverse evaluation scenarios, including cross-domain, unseen domain, and model detection setting. In our main experiment, we use the cross-domain and cross-model setting, which includes various domains and models in both the training and testing set. For the generalizability validation experiments, we utilize the Unseen Domains and Unseen Models of DeepFake, where the former setting means there are unseen-domain texts in the testing set and the latter setting indicates that there are texts generated by unseen LLMs during testing.

- **M4**. The M4 dataset is a comprehensive benchmark spanning multiple domains, models, and languages, comprising data from 8 large language models (LLMs), 6 domains, and 9 languages. It includes human-written content sourced from platforms such as Wikipedia, WikiHow [76], Reddit, arXiv, and PeerRead[77]. Leveraging human-authored prompts, models including ChatGPT [1], DaVinci-003, LLaMA [78], FLAN-T5 [79], Cohere [80] and so on generate outputs across nine languages, including English, Chinese, and Russian. As part of the M4 initiative, a competition [81] is organized to evaluate the detection of AI-generated text at both the paragraph and sentence levels. Two evaluation settings are used: monolingual and multilingual. The multilingual setting introduces new languages absent from the training and validation sets, with AI-generated texts also undergoing paraphrasing. We focus on multilingual setting since this is a more complicated and hard setting for correct LLM detection. The multilingual settings includes 157K training and 42K testing data. The test dataset of M4 also includes the unseen models from the training set.

- **RAID**. RAID[73] is the largest and most comprehensive dataset available for evaluating AI-generated text detection systems. It comprises over 10 million documents, encompassing 11 large language models (LLMs), 11 content genres, 4 decoding strategies, and 12 types of adversarial attacks. The detailed information of RAID dataset can be found in Table 4. We hold out 10% of training data as validation set for evaluation, which includes the texts with all kinds of attacks. We utilize the RAID dataset to compare the robustness of different methods on the attacked texts. During training, we filter the attacked samples in the training set to train the model.

| Category | Values |
|---|---|
| **Models** | ChatGPT, GPT-4, GPT-3 (text-davinci-003), GPT-2 XL, Llama 2 70B (Chat), Cohere, Cohere (Chat), MPT-30B, MPT-30B (Chat), Mistral 7B, Mistral 7B (Chat) |
| **Domains** | ArXiv Abstracts, Recipes, Reddit Posts, Book Summaries, NYT News Articles, Poetry, IMDb Movie Reviews, Wikipedia, Czech News, German News, Python Code |
| **Decoding Strategies** | Greedy (T=0), Sampling (T=1), Greedy + Repetition Penalty (T=0, $\theta$=1.2), Sampling + Repetition Penalty (T=1, $\theta$=1.2) |
| **Adversarial Attacks** | Article Deletion, Homoglyph, Number Swap, Paraphrase, Synonym Swap, Misspelling, Whitespace Addition, Upper-Lower Swap, Zero-Width Space, Insert Paragraphs, Alternative Spelling |

Table 4: RAID dataset composition. The table includes the information about the models, domains, decoding strategies, and adversarial attack kinds.

## B.2   Baseline Setup

In this section, we discuss the details of the experiments setup on baseline models including zero-shot and training-based baselines. For DetectLLM, DetectGPT, DNA-GPT and FastDetectGPT, we

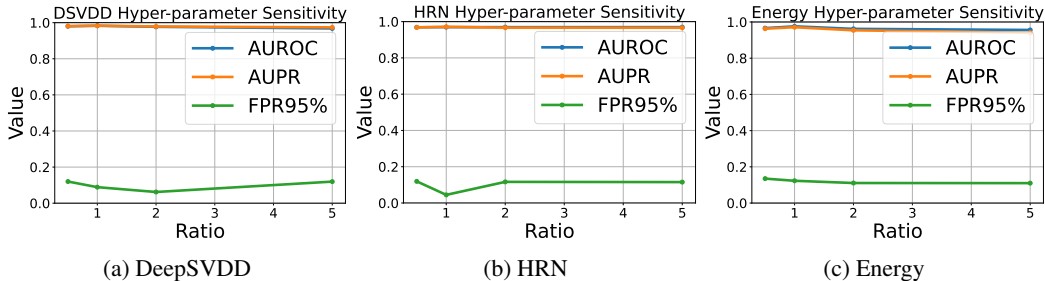

(a) DeepSVDD          (b) HRN          (c) Energy

Figure 3: Hyper-parameter sensitivity analysis of our method. The experiments are conducted on DeepFake dataset and show that our method is robust to the choice of weight.

utilize the official implementation of FastDectGPT[2], which also includes the implementation of other methods. We utilize GPT-Neo-2.7B as the scoring model for all methods. T5-Small is used as the sampling model to do perturbation for DetectGPT and DetectLLM. We do 100 perturbation for each text. Due to the low speed of perturbation, we randomly select 10K samples from each benchmark for evalution for DetectLLM, DetectGPT and DNA-GPT. For DALD, we use the LoRA model trained by GPT-4 texts (which is demonstrated with the best generalizability in the original paper) based on Llama-2-7B as the scoring model. We follow the same experimental setting of Binoculars and apply the Falcon model to compute the final metric. For Glimpse, we utilize the official implementation and call OpenAI davinci-002 API to compute the geometric metric.

For training-based method, such as GPT-Zero, we utilize the open-source version for evaluation[3]. For RADAR and GhostBuster, we apply their official implementation for testing with the updated OpenAI API model. Moreover, we adopt the official open-source implementation of BiScope and utilize gemma-2b as the scoring model.

### B.3 Implementation Detail

We adopt three OOD detection methods including DeepSVDD, HRN and Energy-based methods for our experiments. For DeepSVDD, we use the machine texts from the training set to compute the initial center and compute the corresponding loss with the center. We freeze the parameters of the center point and disable the optimization on the center point. During inference, we compute the $L_2$ distance of the given sample and the center point as the probability of being human-written text. For HRN, follow its original setting by training a one-class classifier per model family using its corresponding data as the positive class and averaging their scores at inference time. We also follow the original hyperparameter settings, where $\lambda = 0.1$ and $n = 12$. For energy-based method, a classification head is attached following the backbone model. We follow [18] to choose hyper-parameters, where $m_{in} = -27$ and $m_{out} = -5$. DeepSVDD is trained from scratch, and HRN and Energy load the pre-trained weights of DeTeCtive. The learning-rate is set as 2e-5 with batch size 32 per device (64 global batch size in our experiments), and the optimizer is Adam [75] with $\beta_1 = 0.9$ and $\beta_2 = 0.98$. The loss weights $\alpha$ and $\beta$ are set as 1 in our experiments. We train all model for 20 epochs. All experiments are conducted on 2*A100 GPUs.

## C More Experimental Results

### C.1 Hyper-parameter Sensitivity Analysis

For the hyperparameter introduced by the DeepSVDD, HRN and Energy-based method, we use the default setting from their original paper. In our work, the hyperparameter we introduce is the ratio of contrastive loss and OOD detection loss, namely $\alpha : \beta$. This ratio balances the contrastive loss and the OOD loss. We conduct a set of experiments with different the loss ratio settings on DeepFake

---

[2]https://github.com/baoguangsheng/fast-detect-gpt
[3]https://github.com/BurhanUlTayyab/GPTZero

Table 5: Performance comparison with the baseline on more evaluation metrics including accuracy and F1 score.

| Method | RAID | |
| | Accuracy ↑ | F1↑ |
| --- | --- | --- |
| DeTeCtive | 96.5 | 55.2 |
| Ours (DSVDD) | 98.6 | 70.0 |
| Ours (HRN) | 98.5 | 67.7 |
| Ours (Energy) | **98.7** | **99.4** |

Table 6: Different backbone model.

| Backbone Model | AUROC ↑ | AUPR↑ | FPR95↓ |
| --- | --- | --- | --- |
| $BERT_{base}$ | 96.9 | 96.7 | 13.0 |
| $BERT_{large}$ | 93.1 | 94.1 | 45.3 |
| $RoBERTa_{base}$ | 96.1 | 96.5 | 16.1 |
| $FLAN\text{-}T5_{base}$ | 95.3 | 95.6 | 18.9 |
| $FLAN\text{-}T5_{large}$ | 96.9 | 96.5 | 9.7 |
| $SimCSE\text{-}BERT_{base}$ | 96.8 | 96.9 | 17.7 |
| $SimCSE\text{-}RoBERTa_{base}$ | 98.3 | 98.3 | 8.9 |
| E5-small | 96.7 | 97.5 | 11.4 |
| E5-multilingual | 96.6 | 96.6 | 12.5 |

dataset and the results are shown Figure 3. Overall, the results are stable and robust to the choice of weight. Ratio 1 : 1 is a great choice which balanced well for all methods and metrics.

## C.2 More Metrics

Besides the AUROC, AUPR and FPR at TPR 95%, we also provide other metrics including the accuracy and F1 score for the comparison. We conduct the experiments on RAID dataset and report the results in Table 5. All settings of our method achieve better performance than the baseline on both accuracy and F1. Our energy-based method shows significant improvement, obtaining 98.7 and 99.4 on accuracy and F1, respectively.

## C.3 Different Backbones

To show the affection of the backbone model for the LLM detection performance, we conduct a further experimental analysis with different training backbones on DeepFake dataset, as shown in Table 6. We can observe that our method is robust to the choice of model backbone. All models generate reasonable results which high AUROC, AUPR and low FRP values.

## C.4 Results on Different Domains

We provide the results of model trained within specific domain on DeepFake dataset in DeepSVDD setting, as shown in Table 7. These strong per-domain performances indicate that the model effectively distinguishes LLM vs. human text within individual domains, further supporting the validity of our OOD formulation.

## C.5 Results on Newer Models

We split the RAID test set to isolate newer models such as GPT-4, ChatGPT, and LLaMA-70B-Chat, and additionally evaluated our method on a recent benchmark, EvoBench [82], using the XSum dataset with GPT-4o outputs, as shown in Table 8. We apply our DeepSVDD setting to train the model on RAID dataset and evaluate directly on RAID subset and GPT-4o set. These results show that our method maintains strong performance even on outputs from state-of-the-art models, demonstrating its effectiveness beyond older-generation LLMs.

Table 7: Results on different domains.

| Domain | AUROC ↑ | AUPR↑ | FPR95↓ |
|--------|---------|-------|--------|
| cmv | 97.9 | 98.3 | 6.7 |
| eli5 | 97.1 | 97.7 | 12.9 |
| hswag | 98.3 | 97.8 | 5.0 |
| roct | 97.6 | 96.5 | 6.0 |
| scigen | 98.7 | 98.3 | 4.0 |
| squad | 99.3 | 99.2 | 2.9 |
| tldr | 97.7 | 97.7 | 8.4 |
| wp | 99.5 | 99.5 | 0.4 |
| xsum | 97.7 | 98.0 | 7.0 |
| yelp | 97.2 | 97.0 | 10.6 |

Table 8: Results on newer models.

| Methods | AUROC ↑ | AUPR↑ | FPR95↓ |
|---------|---------|-------|--------|
| DeTeCtive (RAID subset) | 87.2 | 83.2 | 79.7 |
| Ours (RAID subset) | 93.2 | 74.7 | 39.1 |
| DeTeCtive (GPT4o) | 74.7 | 87.3 | 90.2 |
| Ours (GPT4o) | 100.0 | 100.0 | 0.0 |

## C.6  Results of LLM-generated Text as OOD Sample

Table 9: Results of LLM-generated Text as OOD Sample.

| Methods | AUROC ↑ | AUPR↑ | FPR95↓ |
|---------|---------|-------|--------|
| LLM as OOD | 69.6 | 67.7 | 79.1 |
| Human as OOD | 93.8 | 94.1 | 33.7 |

We provide the result of setting machine-generated text as OOD, as shown in Table 9. The results are trained on DeepSVDD and Deepfake dataset without contrastive loss. The strong performance of human-generated text as OOD indicates that modeling human text as OOD is a valid and effective formulation.

## C.7  Effects of $\alpha$ and $\beta$

We will provide the results when $\alpha$ and $\beta$ are set to 0 in DeepSVDD setting, as shown in Table 10. Without OOD loss ($\beta = 0$), the AUROC is degraded to 88.6 while with only OOD loss, the AUROC achieves 93.7 and the FPR at 95% obtains 33.7, demonstrating the importance of OOD loss.

## C.8  Training and Inference Cost

We provide the training and inference cost comparison of our method and baselines, as shown in Table 11. The time of training and inference is evaluated on single 4090 GPU. The backbone we use is unsup-simcse-roberta-base. For the training time, the training time per epoch is similar to the DeTeCtive baseline since we share the same backbone model and the difference is the loss objective. However, our method leads to faster convergence, which reduces the overall training cost compared with DeTeCtive. For the inference, our method is also the fastest since FastDetectGPT requires LLM inference (such as GPT-Neo-2.7B), which needs 10 × more computation and DeTeCtive needs database construction and KNN for searching, which brings extra inference time.

## C.9  Limitation and Ethical Statement

In this paper, we theoretically and empirically analyze why and how to model LLM text detection task as the out-of-distribution detection task. However, current detection can only tell whether the given text is AI-generated or human-written, which lacks further interpretation about how the text is

Table 10: Effects of $\alpha$ and $\beta$.

| Parameter Setting | AUROC $\uparrow$ | AUPR$\uparrow$ | FPR95$\downarrow$ |
|---|---|---|---|
| $\beta = 0$ | 88.6 | 93.5 | 76.1 |
| $\alpha = 0$ | 93.7 | 94.1 | 33.7 |
| $\alpha = 1, \beta = 1$ | 98.3 | 98.3 | 8.9 |

Table 11: Training and inference cost of different methods.

| Method | Training Time per Epoch | Overall Training Time | Inference time |
|---|---|---|---|
| FastDetectGPT | N/A | N/A | 72 min |
| DeTeCtive | 0.67 h | 20 h (30 epoch) | 29 min |
| Ours | 0.67 h | 3.3 h (5 epoch) | 4 min |

generated by LLM. Therefore, we hope the user uses the detection result as a reference and doesn't make decisions solely based on the detection results, especially in areas such as academia.

