# OpenReview forum: "Human Texts Are Outliers: Detecting LLM-generated Texts via Out-of-distribution Detection"
_NeurIPS.cc/2025/Conference — NeurIPS 2025 poster_

### Official Review · Reviewer_vd2x · 2025-06-30

**Clarity:** 2
**Significance:** 2
**Originality:** 3
**Rating:** 4
**Confidence:** 4

**Summary:**

The paper proposes to reformulate the problem of detecting LLM-generated text from binary classification to an out-of-distribution (OOD) detection task. The authors argue that human-written texts are inherently diverse and cannot be modeled as a single distribution, while machine-generated texts exhibit more stable statistical properties. They use one-class (DeepSVDD, HRN) and score-based (energy-based) OOD detection methods to detect human texts as outliers. Theoretical analysis supports the failure of binary classifiers due to distribution mismatch, and extensive experiments on multilingual, adversarial, and unseen-domain settings demonstrate strong empirical performance, outperforming prior zero-shot and training-based baselines.

**Questions:**

- **(Q1)** Your approach relies on score-based decisions (e.g., DeepSVDD radius, HRN sigmoid, energy scores), but the paper does not describe how inference thresholds are selected. Are these thresholds fixed, learned, or tuned on validation data? Are they consistent across datasets or adjusted per domain? Please clarify this, and ideally include ablation or robustness analysis over threshold choices.
- **(Q2)** The proposed framework combines the OOD loss with a supervised contrastive loss. However, your ablation study (Table 3) only compares different OOD losses against binary classification, but does not isolate the effect of the contrastive term. Could you include an ablation where $\alpha = 0$ and separately $\beta = 0$ to assess how much of the gain comes from contrastive learning? This would help assess whether the gains are due to the OOD formulation itself or enhanced representation learning from contrastive supervision.
- **(Q3)** Can you quantify somehow intra-class variance of LLM texts across different domains or models? It would be better to show statistically that LLM texts are indeed statistically coherent and form a compact in-distribution set. This would strengthen the argument for using a one-class formulation and help validate the theoretical motivation.
- **(Q4)** Most datasets used in evaluation include outputs from relatively early-generation LLMs. Do you plan to test on newer model outputs, or maybe select a subset from data which contains only newer models?

**Ethical Concerns:**

["NO or VERY MINOR ethics concerns only"]

**Final Justification:**

The rebuttal convincingly addressed major concerns, especially through strong ablation results and evaluations on newer LLM outputs, which improved my confidence. The inter-/intra-class analysis and theoretical framing are interesting, though additional quantitative evidence could further prove the need for the OOD formulation. I believe, it is worth improving. As limitation also remains the scope of evaluation

**Limitations:**

Yes

**Quality:**

2

**Strengths And Weaknesses:**

Strengths:
- **(S1)** Conceptual Novelty and Motivation: the paper treats machine-generated text detection as an out-of-distribution (OOD) detection task, rather than a binary classification problem. This is a novel and theoretically motivated perspective
- **(S2)** Theoretical Proofs and Clarity: authors support their claim with solid theoretical analysis, including formal proofs of how binary classifiers fail under realistic distribution shifts. This gives strong credibility to the motivation for the OOD formulation.
- **(S3)** Strong Empirical Results: the proposed framework achieves state-of-the-art results across several benchmarks (DeepFake, M4-multilingual, RAID).

Weaknesses:
- **(W1)** Unvalidated Assumption of LLM Homogeneity: a central assumption is that LLM-generated texts form a compact, learnable distribution, unlike human texts. However, this assumption is neither empirically validated nor tested across different models or domains. If LLM texts are multimodal or diverse, this could possibly make method worse
- **(W2)** Lack of Ablation: the proposed framework combines the OOD loss with a supervised contrastive loss. However, the paper does not ablate the contribution of this component. It is unclear whether the performance gains are because of the OOD formulation itself or primarily from the contrastive learning
- **(W3)** Old Datasets: most of the datasets contains relatively old models, which generations could be very easily detected

---

> ### Author Rebuttal · Authors · 2025-07-31
>
> Thank you for your insightful review. To address your concern, we would like to provide the response to your reviews:
>
> > W1: Unvalidated Assumption of LLM Homogeneity: a central assumption is that LLM-generated texts form a compact, learnable distribution, unlike human texts. However, this assumption is neither empirically validated nor tested across different models or domains. If LLM texts are multimodal or diverse, this could possibly make method worse
>
> Thank you for your suggestion. As NeurIPS does not permit the inclusion of PDFs or anonymous external links in the rebuttal this year, we apologize in advance that we are unable to provide annotated figures or visual clarifications. We hope our written explanations are sufficiently clear, and we are happy to incorporate this additional clarifications in the final version. For this question, we can provide the inter- and intra-LLM and human text cosine distance across diverse domains:
>
> | **Metric**           | **Dist** |
> |----------------------|----------|
> | Intra-distance LLM   |   0.3014 |
> | Intra-distance human |   0.4747 |
> | Inter-distance       |   1.6048 |
>
> We compute the average cosine distance on the Deepfake dataset, which contains domains such as cmv, xsum, eli5 and so on, and LLMs including GPT-3.5-turbo, GLM-130B, Bloom-7B and so on. The embedding backbone we use is RoBERTa pretrained with contrastive loss. The results show that the Intra-distance of LLM is smaller than the human while the inter-distance between human and LLM is relatively large, supporting our OOD hypothesis that LLM text forms a coherent in-distribution cluster, while human-written texts—spanning varied styles and domains—act as natural OOD examples.. We hope this can address your concern.
>
>
> > W2: Lack of Ablation: the proposed framework combines the OOD loss with a supervised contrastive loss. However, the paper does not ablate the contribution of this component. It is unclear whether the performance gains are because of the OOD formulation itself or primarily from the contrastive learning
>
> Thank you for your suggestion. We will the ablation of each loss component:
>
> | **Methods**             | **AUC_ROC** | **PR_AUC** | **FPR-95%** |
> |-------------------------|:-----------:|:----------:|:-----------:|
> | Contrastive Loss        |        88.6 |       93.5 |        76.1 |
> | OOD loss                |        93.7 |       94.1 |        33.7 |
> | Contrastive + OOD Loss  |        98.3 |       98.3 |         8.9 |
>
> These results highlight the critical role of the OOD loss. Even without contrastive learning, the OOD loss alone substantially improves performance, especially in terms of FPR-95%, indicating its strong discriminative power. While contrastive learning offers additional benefits, the OOD loss is the primary driver of robustness and separation between distributions.
>
> > W3: Old Datasets: most of the datasets contains relatively old models, which generations could be very easily detected
>
> Thank you for the helpful comment. At the time of our submission, RAID was already the most up-to-date benchmark available. To further address your concern, we split the RAID test set to isolate newer models such as GPT-4, ChatGPT, and LLaMA-70B-Chat, and additionally evaluated our method on a recent benchmark, EvoBench [1], using the XSum dataset with GPT-4o outputs. The results are as follows:
>
> | **Newer Model only**    | **AUC_ROC** | **PR_AUC** | **FPR-95%** |
> |-------------------------|:-----------:|:----------:|:-----------:|
> | DeTeCtive (RAID subset) |        87.2 |       83.2 |        79.7 |
> | Ours (RAID subset)      |        93.2 |       74.7 |        39.1 |
> | DeTeCtive (GPT4o)       |        74.7 |       87.3 |        90.2 |
> | Ours (GPT4o)            |       100.0 |      100.0 |         0.0 |
>
> These results show that our method maintains strong performance even on outputs from state-of-the-art models, demonstrating its effectiveness beyond older-generation LLMs.
>
>
> > Q1: Your approach relies on score-based decisions (e.g., DeepSVDD radius, HRN sigmoid, energy scores), but the paper does not describe how inference thresholds are selected. Are these thresholds fixed, learned, or tuned on validation data? Are they consistent across datasets or adjusted per domain? Please clarify this, and ideally include ablation or robustness analysis over threshold choices.
>
> Thank you for the thoughtful question. We would like to clarify that our evaluation metrics—ROC-AUC, PR-AUC, and FPR at 95% TPR—are all threshold-independent or use standardized criteria for threshold selection. Specifically, ROC-AUC and PR-AUC measure performance across all possible decision thresholds, providing a holistic view of the model's discriminative ability. The FPR at 95% TPR metric, while threshold-based, uses a fixed target TPR to compute the corresponding FPR, ensuring consistency and comparability across models and datasets. Therefore, our method does not require the manual tuning or learning of specific thresholds during inference, and the reported results are not sensitive to any particular threshold choice.
>
> > Q2: The proposed framework combines the OOD loss with a supervised contrastive loss. However, your ablation study (Table 3) only compares different OOD losses against binary classification, but does not isolate the effect of the contrastive term. Could you include an ablation where and separately to assess how much of the gain comes from contrastive learning? This would help assess whether the gains are due to the OOD formulation itself or enhanced representation learning from contrastive supervision.
>
> Thank you for your question. We have already provided the ablation study of each loss component in W2. Please have a check. Hope this could address your concern.
>
> > Q3: Can you quantify somehow intra-class variance of LLM texts across different domains or models? It would be better to show statistically that LLM texts are indeed statistically coherent and form a compact in-distribution set. This would strengthen the argument for using a one-class formulation and help validate the theoretical motivation.
>
> Yes, we have already provided the intra and inter distance in W1. Please have a check.
>
> > Q4: Most datasets used in evaluation include outputs from relatively early-generation LLMs. Do you plan to test on newer model outputs, or maybe select a subset from data which contains only newer models?
>
> Yes, we have provided the results in W3. Please have a check.
>
>
> [1] Yu, Xiao, et al. "EvoBench: Towards Real-world LLM-Generated Text Detection Benchmarking for Evolving Large Language Models." Findings of the Association for Computational Linguistics: ACL 2025. 2025.
>
> We hope our responses adequately address all your concerns. If you have any further questions or feedback, we would be glad to continue the discussion during the discussion phase.

---

> > ### Comment · Reviewer_vd2x · 2025-08-03
> >
> > Thank you for the detailed rebuttal and additional experiments. The ablation study is very helpful in clarifying the contributions of the OOD loss versus the contrastive term, and the strong results on newer models like GPT-4 and GPT-4o help address concerns about outdated datasets. Many of my concerns have been addressed.
> >
> > The inter- and intra-class cosine distance analysis is interesting. However, if LLM- and human-generated texts lie in distinct regions (low intra-class and high inter-class cosine distances), doesn’t this suggest that the data could be separated by a simple hyperplane? Also, were the features L2-normalized before computing cosine distances? If not, could you please provide results with normalization?
> >
> > Thanks again for the thoughtful clarifications.

---

> ### Author Response · Authors · 2025-08-04
> **Response to the follow-up**
>
> Thank you for your insightful follow-up questions.
>
> You are absolutely right in noting that the distinct clustering of LLM- and human-generated texts suggests the feasibility of separating them using a simple hyperplane. In fact, as shown in Table 3, even a basic classification head achieves reasonable  performance across all benchmarks (e.g., AUROC of 89.0 on DeepFake). However, we emphasize that merely relying on a classification head may limit generalization, especially in scenarios such as higher-quality LLMs and adversarial paraphrasing (as you can see, the performance of the single classification head drops on the RAID dataset, which includes various attacked data), where the boundary between LLM and human texts becomes more subtle. Therefore, instead of learning to discriminate human versus LLM texts directly, our method focuses on learning a compact and consistent representation of LLM outputs, treating diverse human-written texts as OOD due to the difficulty of modeling true human distribution. Moreover, when computing the distance, we have already applied the L2-normalization to the output representation.
>
> We hope this can address your concerns. Thanks a lot.

---

> > ### Comment · Reviewer_vd2x · 2025-08-05
> >
> > Thank you for the detailed clarification. It was very helpful for my understanding, and I am raising my score to 4.
> >
> > Regarding the cosine distance analysis, I still believe it would strengthen the argument and show the necessity of the OOD detection formulation, if you could perform more comprehensive analysis. For example, you could evaluate adversarial paraphrasing or split AI-generated text by LLM type and compute cosine similarities, I believe it could give some insights.

---

> > > ### Author Response · Authors · 2025-08-07
> > > **Response to Reviewer vd2x**
> > >
> > > Thank you very much for your thoughtful engagement. We're also grateful for your continued suggestions regarding the cosine distance analysis. We provide the distance analysis of the subset of RAID dataset, which includes adversarial paraphrasing:
> > >
> > > | **Metric**           | **Dist** |
> > > |----------------------|----------|
> > > | Intra-distance LLM   |   0.1929 |
> > > | Intra-distance human |   0.8698 |
> > > | Inter-distance       |   0.7859 |
> > >
> > > We observe that the overall pattern remains consistent, but the Inter-distance is relatively smaller, indicating that it becomes more difficult for a simple classifier to distinguish between the two.
> > >
> > > We will include all of the distance analyses in our revision. Thank you very much for your constructive feedback that helped strengthen our work.

---

### Official Review · Reviewer_twCw · 2025-07-02

**Clarity:** 3
**Significance:** 3
**Originality:** 2
**Rating:** 3
**Confidence:** 4

**Summary:**

The paper argues that treating AI-text detection as a binary human vs. machine task is theoretically unsound because human writing is an open-world, highly heterogeneous distribution. It therefore reframes the problem as out-of-distribution (OOD) detection: model only the in-distribution (ID) machine-generated manifold and flag any deviation as human-authored. Concretely, the authors instantiate three OOD objectives—DeepSVDD, HRN, and Energy-based scoring—on top of a RoBERTa encoder and combine them with a supervised contrastive term. Extensive experiments on DeepFake, M4-multilingual and RAID show large gains (e.g. 98.3 % AUROC, 8.9 % FPR95 on DeepFake) over zero-shot and supervised baselines, with additional analyses for multilingual, adversarial and unseen-domain/model settings. Practical guidance on when to choose each loss is provided.

**Questions:**

See Weaknesses.

**Ethical Concerns:**

["NO or VERY MINOR ethics concerns only"]

**Final Justification:**

My major concern is the limited novelty. After reading the manuscript, I found this paper is essentially a incremental or combinational work, where all the parts of the proposed method originates from the estabilished literature without any specific modification tailor to LLM-generated text detection. Even if considering this paper as a benchmarking paper, it seems insufficient for the authors to only evaluate two OOD detection losses. This raises several questions that is unanswered in the manuscript:

what is the motivation of the authors use the two losses?
how other OOD detection losses performs in LLM-generated text detection. if they fail to work, why?

**Quality:**

2

**Strengths And Weaknesses:**

Strengths:
1. Clear theoretical motivation. Theorem 2 formalises why binary classifiers overfit when human heterogeneity is large.
2. Casting detection as one-class/OOD aligns with the asymmetric data geometry illustrated in Fig. 1 and avoids modelling an unbounded human space.
3. All three OOD losses surpass prior detectors across AUROC, AUPR and FPR95 on three benchmarks.

Weaknesses:
1. One-class/OOD ideas have a long history in NLP anomaly detection; related work could cite and empirically compare to prior one-class text detectors.
2. The χ²-divergence in Theorem 2 is not empirically estimated, so the link between theory and practice remains qualitative.
3. Encoder choice fixed to RoBERTa. How would results change with more recent lightweight or multilingual encoders?
4. How sensitive are DeepSVDD radius or energy margins to training-set size?

---

> ### Author Rebuttal · Authors · 2025-07-31
>
> Thank you for your detailed review and comment. To address your concern, we would like to provide the response to your reviews:
>
> > W1: One-class/OOD ideas have a long history in NLP anomaly detection; related work could cite and empirically compare to prior one-class text detectors.
>
> We believe there is OOD idea in previous NLP anomaly detection for classification, which is not applied in LLM detection. However, we would like to reiterate that we are the first to apply out-of-distribution detection methods to LLM detection. Moreover, we validate our method empirically and theoretically.  Specifically, we have demonstrated the effectiveness of the algorithm on a range of classic OOD methods such as DeepSVDD, Energy-based and HRN OOD detection methods. Our method achieves SoTA performance on diverse LLM detection tasks including multilingual and attacked text settings. Meanwhile, the generalizability is also validated.
>
> We would like to kindly ask the reviewer to provide the specific citation of prior one-class text detectors. We are happy to provide the comparison in LLM detection settings.
>
> > W2: The χ²-divergence in Theorem 2 is not empirically estimated, so the link between theory and practice remains qualitative.
>
> Thank you for your suggestion. We would like to provide the empirical  $\chi^2$-divergence estimation of our setting:
>
> | **Domain**    | **Divergence** |
> |---------------|----------------|
> | wp_human      |         1.8790 |
> | hswag_human   |         0.3229 |
> | yelp_human    |         0.3107 |
> | sci_gen_human |        24.6070 |
> | squad_human   |         0.1500 |
> | eli5_human    |        22.1010 |
> | tldr_human    |         0.2894 |
> | xsum_human    |         0.1820 |
> | cmv_human     |         0.6971 |
> | roct_human    |         1.0137 |
> | Avg.          |         9.0491 |
>
> We evaluate the $\chi^2$-divergence on Deepfake dataset, which contains multiple domains and source LLMs. We use the full human set as the true distribution and sample subsets as the fitted distribution. We apply the RoBERTa model to compute the embedding. The results show that while some domains such as squad and xsum obtain low $\chi^2$-divergence, some other domains like sci_gen and eli5 gain significantly high divergence. Moreover, the average divergence is also large (9.0491), demonstrating the gap between the fitted and true human distribution.
>
> We would like to point out that in Theorem 2, we unfortunately have a typo: the last term for the loss of $\hat{D}$ should be $0.9 q_H (D+1) \cdot \Delta$ instead of $0.9 q_H (D-1) \cdot \Delta$. We corrected this typo in the appendix and our final version. Thus even with smaller values of the $\chi^2$-divergence (e.g. D<1), the potential difference in loss is still significant.
>
> > W3: Encoder choice fixed to RoBERTa. How would results change with more recent lightweight or multilingual encoders?
>
> Thank you for your suggestion. In fact, we already included the results of different backbones in Table 6 in the appendix including BERT, RoBERTa and FLAN-T5. Following your suggestion, we provide results of more efficient backbone E5-small with 33M parameters and Multilingual-E5-base:
>
> | **Domain**      | **AUC_ROC** | **PR_AUC** | **FPR-95%** |
> |-----------------|:-----------:|:----------:|:-----------:|
> | E5-small        |        96.7 |       97.5 |        11.4 |
> | E5-multilingual |        96.6 |       96.6 |        12.5 |
>
> The model is trained with the identical setting on DeepFake and DeepSVDD. Both efficient and multilingual backbones achieve comparable results, indicating the robustness of our method for backbone choice.
>
>
> > W4: How sensitive are DeepSVDD radius or energy margins to training-set size?
>
> We clarify that in our DeepSVDD experiments, we only use simple one-class setting which does not rely on the radius. For the energy margins and training-set size, we provide the sensitivity analysis experiment on DeepFake dataset:
>
> | **Setting**                   | **AUC_ROC** | **PR_AUC** | **FPR-95%** |
> |-------------------------------|:-----------:|:----------:|:-----------:|
> | M_in=-27, M_out=-2, full set  |        97.6 |       97.2 |        12.3 |
> | M_in=-30, M_out=-2, full set  |        96.7 |       96.0 |         7.4 |
> | M_in=-20, M_out=-2, full set  |        96.7 |       95.9 |         7.5 |
> | M_in=-27,  M_out=-5, full set |        97.1 |       96.5 |         6.1 |
> | M_in=-27, M_out=0, full set   |        96.4 |       95.5 |         8.8 |
> | M_in=-27, M_out=-2, half set  |        97.1 |       96.4 |         6.2 |
> | M_in=-30, M_out=-2, half set  |        97.1 |       96.5 |         6.7 |
> | M_in=-20, M_out=-2, half set  |        96.6 |       95.7 |         8.0 |
> | M_in=-27,  M_out=-5, half set |        97.1 |       96.4 |         6.2 |
> | M_in=-27, M_out=0, half set   |        96.3 |       95.4 |         8.7 |
>
> We can see that the model performance is robust with different energy margin and training set size.
>
> We hope our responses adequately address all your concerns. If you have any further questions or feedback, we would be glad to continue the discussion during the discussion phase.

---

### Official Review · Reviewer_VQzR · 2025-07-03

**Clarity:** 3
**Significance:** 2
**Originality:** 3
**Rating:** 4
**Confidence:** 3

**Summary:**

This paper reframes the task of distinguishing human-written from machine-generated text as an out-of-distribution (OOD) detection problem rather than a binary classification task. This is motivated by the theory that the human text distribution is not simply a single distribution due to the way that linguistics works. The authors argue that instead, the LLM distribution should be viewed as a single distribution, with human text distributions being "out-of-distribution". This leads to a novel framework for LLM text detection based on OOD detection methods, and a supervised method is developed. Experimental results on a suite of benchmark LLM text detection datasets is provided to support the method.

**Questions:**

See weaknesses.

**Ethical Concerns:**

["NO or VERY MINOR ethics concerns only"]

**Final Justification:**

The authors have addressed my overall concerns, and I will keep my positive score.

**Limitations:**

yes

**Quality:**

3

**Strengths And Weaknesses:**

Strengths:
- Reformulating LLM text detection as OOD rather than binary classification is a compelling conceptual shift that aims to provide a more fine-grained assumption on human and LLM text distributions. Prior works may be too simplistic in their assumptions by assuming that there is an all-encompassing human text distribution and corresponding LLM text distribution.
- the performance is good on the benchmarks provided, which are extensive. These help support that the OOD framework and method are effective and successful in solving the OOD problem.

Weaknesses:
- While the authors provide experiments supporting their method, some of the intuition provided could benefit from some experiments that demonstrate the validity of the assumptions. For example, the OOD framework and modeling the LLM text distribution as "in-distribution" could benefit from some plots demonstrating that this is indeed the case in practice. My understanding is that Figure 1 is not a visualization of text embeddings on real data. If a figure demonstrating, say, the text embeddings of LLM texts and different human texts (say, coming from different domains or writing patterns) was shown, this would help support the OOD hypothesis. This could help clarify whether textual domain shift or human/LLM writing is the primary cause for a distribution shift. Right now it seems the authors argue that all LLM text across various domains can be viewed as in-distribution, whereas human text distributions are more varied.
- Building on the previous point, I think some experiments regarding domain vs. human/LLM should be provided. My understanding is that the LLM-written parts of the training splits of the datasets are used to train the models. But if further splits across domains could be provided, that would be helpful to understand the method.
- Is there a zero-shot variant based on the OOD framework that could be developed? The training of such detectors can present obstacles to future times when stronger and higher-performing LLMs are developed.
- A brief comparison of training/inference time or parameter counts versus baselines would be helpful to understand there performance/complexity tradeoffs, which is another practical concern when deploying such detectors.
- Recent focus has shifted to edited LLM texts, or partial human/LLM text. This is a more accurate representation of how LLM texts are used in practice. It would be interesting to see how the method would work on such data.

---

> ### Author Rebuttal · Authors · 2025-07-31
>
> Thank you for your concrete comment and suggestion. To address your concern, we would like to provide the response to your reviews:
>
> > W1: While the authors provide experiments supporting their method, some of the intuition provided could benefit from some experiments that demonstrate the validity of the assumptions. For example, the OOD framework and modeling the LLM text distribution as "in-distribution" could benefit from some plots demonstrating that this is indeed the case in practice. My understanding is that Figure 1 is not a visualization of text embeddings on real data. If a figure demonstrating, say, the text embeddings of LLM texts and different human texts (say, coming from different domains or writing patterns) was shown, this would help support the OOD hypothesis. This could help clarify whether textual domain shift or human/LLM writing is the primary cause for a distribution shift. Right now it seems the authors argue that all LLM text across various domains can be viewed as in-distribution, whereas human text distributions are more varied.
>
> As NeurIPS does not permit the inclusion of PDFs or anonymous external links in the rebuttal this year, we apologize in advance that we are unable to provide annotated figures or visual (such as t-SNE of embeddings) clarifications. We hope our written explanations are sufficiently clear, and we are happy to incorporate this additional clarification in the final version. For this question, we can provide the inter- and intra-LLM and human text cosine distance across diverse domains:
>
> | **Metric**           | **Dist** |
> |----------------------|----------|
> | Intra-distance LLM   |   0.3014 |
> | Intra-distance human |   0.4747 |
> | Inter-distance       |   1.6048 |
>
> We compute the average cosine distance on the Deepfake dataset, which contains domains such as cmv, xsum, eli5 and so on, and LLMs including GPT-3.5-turbo, GLM-130B, Bloom-7B and so on. The embedding backbone we use is RoBERTa pretrained with contrastive loss. The results show that the Intra-distance of LLM is smaller than the human while the inter-distance between human and LLM is relatively large, supporting our OOD hypothesis that LLM text forms a coherent in-distribution cluster, while human-written texts—spanning varied styles and domains—act as natural OOD examples. We hope this can address your concern.
>
>
> > W2: Building on the previous point, I think some experiments regarding domain vs. human/LLM should be provided. My understanding is that the LLM-written parts of the training splits of the datasets are used to train the models. But if further splits across domains could be provided, that would be helpful to understand the method.
>
> Yes, thanks for the suggestion, we provide the results of model trained within specific domain:
>
> | **Domain** | **AUC_ROC** | **PR_AUC** | **FPR-95%** |
> |------------|:-----------:|:----------:|:-----------:|
> | cmv        |        97.9 |       98.3 |         6.7 |
> | eli5       |        97.1 |       97.7 |        12.9 |
> | hswag      |        98.3 |       97.8 |         5.0 |
> | roct       |        97.6 |       96.5 |         6.0 |
> | sci_gen    |        98.7 |       98.3 |         4.0 |
> | squad      |        99.3 |       99.2 |         2.9 |
> | tldr       |        97.7 |       97.7 |         8.4 |
> | wp         |        99.5 |       99.5 |         0.4 |
> | xsum       |        97.7 |       98.0 |         7.0 |
> | yelp       |        97.2 |       97.0 |        10.6 |
>
> These strong per-domain performances indicate that the model effectively distinguishes LLM vs. human text within individual domains, further supporting the validity of our OOD formulation.
>
> > W3: Is there a zero-shot variant based on the OOD framework that could be developed? The training of such detectors can present obstacles to future times when stronger and higher-performing LLMs are developed
>
> We would like to first thank the reviewer for providing such an insightful and constructive proposal. It is indeed pretty interesting to extend our out-of-distribution detection framework to zero-shot methods. However, it is non-trivial and challenging to simply apply our current framework to zero-shot methods. The reason is that the current OOD detection methods generally require supervised (one-class) training, which is not compatible with current metric-based zero-shot methods. Nevertheless, we believe the idea behind our method, namely considering the human-generated texts as OOD data, can help to design novel statistic metrics for zero-shot methods, for example, help to find the best metric threshold in the zero-shot methods.
>
> > W4: A brief comparison of training/inference time or parameter counts versus baselines would be helpful to understand the performance/complexity tradeoffs, which is another practical concern when deploying such detectors.
>
> Thank you for you suggestion. We provide the training and inference cost comparison of our method and baselines:
>
> | **Method**    | **Training Time per Epoch** | **Overall Training Time** | **Inference time** |
> |---------------|-----------------------------|---------------------------|--------------------|
> | FastDetectGPT | N/A                         | N/A                       | 72min              |
> | DeTeCtive     | 0.67h                       | 20h (30 epoch)            | 29min              |
> | Ours          | 0.67h                       | 3.3h (5 epoch)            | 4min               |
>
> The time of training and inference is evaluated on single 4090 GPU. The backbone we use is  unsup-simcse-roberta-base. For the training time, the training time per epoch is similar to the DeTeCtive baseline since we share the same backbone model and the difference is the loss objective. However, our method leads to faster convergence, which reduces the overall training cost compared with DeTeCtive. For the inference, our method is also the fastest since FastDetectGPT requires LLM inference (such as GPT-Neo-2.7B), which needs 10 $\times$ more computation and DeTeCtive needs database construction and KNN for searching, which brings extra inference time.
>
> > W5: Recent focus has shifted to edited LLM texts, or partial human/LLM text. This is a more accurate representation of how LLM texts are used in practice. It would be interesting to see how the method would work on such data.
>
> Thanks for raising this up. We want to clarify that the test set RAID used in our experiments has already included the attacked data including paraphrase, insert paragraphs and so on. Moreover, we would like to provide the results comparison with baselines on test set with only attacked text from RAID:
>
> | **Attack-only RAID** | **AUC_ROC** | **PR_AUC** | **FPR-95%** |
> |----------------------|:-----------:|:----------:|:-----------:|
> | DeTeCtive            |        81.9 |       72.0 |        86.0 |
> | Ours                 |        96.1 |       70.2 |        19.8 |
>
> Our method achieves a substantially higher AUC_ROC and significantly lower FPR-95% compared to the baseline, indicating strong robustness on edited or partially human-generated text.
>
>
> We hope our responses adequately address all your concerns. If you have any further questions or feedback, we would be glad to continue the discussion during the discussion phase.

---

> > ### Comment · Reviewer_VQzR · 2025-08-07
> >
> > Thank you for the thoughtful response and addressing my concerns. I will keep my positive rating.

---

### Official Review · Reviewer_mbi1 · 2025-07-03

**Clarity:** 3
**Significance:** 3
**Originality:** 3
**Rating:** 4
**Confidence:** 5

**Summary:**

Paper propose a methods for detecting LLM generated text. The proposed approach formulates LLM generated text detection as a single class out-of-distribution detection problem, where human-written texts is treated distributional outliers while machine-generated texts are treated as in-distribution data. Existing work often formulate it is a binary classification problem. Author show theoretically limitations in  binary classifier for human-written text distribution particularly ability to generalize. Proposed approach is implemented using three variants of Single class learning methods. All of them showed better accuracy then using a binary classification head and loss on same network. Proposed approach is benchmarked on 3 different datasets and show better performance. Approach also has ability to generalize to unseen domain and text generated from unseen models.

**Questions:**

1. some theoretical/empirical justification of considering human generated text as OOD v/s machine generated text as OOD will help
2. benchmarking gain from adding OOD loss to contrastive loss or removing contrastive loss completely can inform better if the gains is from single class modeling or improving contrastive loss with another loss term that constraints intra-class embeddings for machine generated text?

**Ethical Concerns:**

["NO or VERY MINOR ethics concerns only"]

**Final Justification:**

Author have addressed both of my concerns, ablation support authors modeling human text as OOD and impact of OOD loss on improving performance. I am more confident with my recommendation now.

**Quality:**

3

**Strengths And Weaknesses:**

Major Strength:
1. Author provide theoretical and empirical confirmation on why binary loss and classification can be incomplete for training a generalizable  LLM generated text detector. Improvements in Table 3, show significant gain w.r.t binary classification.
2. Method if fairly novel. It is common to use one-class detector for detecting machine generated audio, image and video (e.g. R1, R2). However, work is limited on LLM generated text on this. Also, typically existing methods fit bonafide/real media and consider machine generated media as OOD data. However, this approach does it other way around. Which make intuitively some sense specifically for text domain where approaches are mostly autoregressive and based on similar training loss and architecture.
3. Performance is competitive, author compared with enough baselines and ran benchmark generalizability of method.

Major Weakness:
1. No empirical or theoretical evidence is provided about a difference in choice from existing work. That is setting human generated text as OOD and not machine generated text as OOD.
2. While author formulate problem as single class, from equation 6 and equation 10 i.e. L_contrastive loss it seems author use both human and machine generated text during training. Since the loss weights α and β are set as 1 in main experiments, the contrastive loss (binary classification loss) also play equal role in training. Hence, it would be good to understand performance of method with β set to 0 to truly understand if the gains are from single class modeling or augmenting contrastive loss with another loss that constraints intra-class embeddings for machine generated text. In supplement Figure 3, author show metrics remaining same for different ratio of alpha:beta however adding ablation for β =0 and α =0 will help.


R1: https://arxiv.org/pdf/2406.16716 (InterSpeech 2024)
R2: K. Lee, Y. Zhang and Z. Duan, "A Multi-Stream Fusion Approach with One-Class Learning for Audio-Visual Deepfake Detection," 2024 IEEE 26th International Workshop on Multimedia Signal Processing (MMSP), West Lafayette, IN, USA, 2024, pp. 1-6, doi: 10.1109/MMSP61759.2024.10743671

---

> ### Author Rebuttal · Authors · 2025-07-31
>
> Thank you for your detailed review. To address your concern, we would like to provide the response to your reviews:
>
> > W1: No empirical or theoretical evidence is provided about a difference in choice from existing work. That is setting human generated text as OOD and not machine generated text as OOD.
>
> Thanks for the suggestion. We provide the result of setting machine-generated text as OOD:
>
> | **Methods**    | **AUC_ROC** | **PR_AUC** | **FPR-95%** |
> |----------------|:-----------:|:----------:|:-----------:|
> | machine as OOD |        69.6 |       67.7 |        79.1 |
> | human as OOD   |        93.8 |       94.1 |        33.7 |
>
> The results are trained on DeepSVDD and Deepfake dataset without contrastive loss. The strong performance of human generated text as OOD indicates that modeling human text as OOD is a valid and effective formulation.
>
> > W2: While author formulate problem as single class, from equation 6 and equation 10 i.e. L_contrastive loss it seems author use both human and machine generated text during training. Since the loss weights α and β are set as 1 in main experiments, the contrastive loss (binary classification loss) also play equal role in training. Hence, it would be good to understand performance of method with β set to 0 to truly understand if the gains are from single class modeling or augmenting contrastive loss with another loss that constraints intra-class embeddings for machine generated text. In supplement Figure 3, author show metrics remaining same for different ratio of alpha:beta however adding ablation for β =0 and α =0 will help.
>
> We will provide the results when $\alpha$ and $\beta$ is set to 0:
>
> | **Methods**               | **AUC_ROC** | **PR_AUC** | **FPR-95%** |
> |---------------------------|:-----------:|:----------:|:-----------:|
> | $\beta$ = 0               |        88.6 |       93.5 |        76.1 |
> | $\alpha$ = 0              |        93.7 |       94.1 |        33.7 |
> | $\beta$ = 1, $\alpha$ = 1 |        98.3 |       98.3 |         8.9 |
>
> Without OOD loss ($\beta$ = 0), the AUC_ROC is degraded to 88.6 while with only OOD loss, the AUC_ROC achieves 93.7 and the FPR at 95% obtains 33.7, demonstrating the importance of OOD loss.
>
> > Q1: some theoretical/empirical justification of considering human generated text as OOD v/s machine generated text as OOD will help
>
> Thank you for your question. We have already provided the results comparison of  human-generated text as OOD and machine-generated text as OOD in W1. Please have a check.
>
> > Q2: benchmarking gain from adding OOD loss to contrastive loss or removing contrastive loss completely can inform better if the gains is from single class modeling or improving contrastive loss with another loss term that constraints intra-class embeddings for machine generated text?
>
> We have provided the results of different loss choice in W2. Please have a check.
>
> We hope our responses adequately address all your concerns. If you have any further questions or feedback, we would be glad to continue the discussion during the discussion phase.

---

> ### Comment · Reviewer_mbi1 · 2025-08-06
>
> I thank the authors for running additional experiments to address my concerns. My concerns regarding human as OOD v/s machine as OOD and role of OOD loss in performance has been resolved. I am more confident about my positive recommendation now. Thanks

---

> > ### Author Response · Authors · 2025-08-07
> >
> > Thank you again for your detailed review and insightful feedback, which have been instrumental in improving the quality of our work.

---

### Decision · Program_Chairs · 2025-09-17

**Decision:**

Accept (poster)

**Comment:**

The proposed approach formulates LLM generated text detection as a single class out-of-distribution detection problem, where human-written texts is treated distributional outliers while machine-generated texts are treated as in-distribution data.

The strength of the paper lies in reformulating LLM text detection as OOD rather than binary classification is a compelling conceptual shift that aims to provide a more fine-grained assumption on human and LLM text distributions, and its extensive experiments.

For weakness, the AC summarize the discussion between the author and reviewers, where almost all weaknesses are properly addressed:
1. Justification and Validation of the OOD Assumption
Reviewers' Points (mbi1, VQzR, vd2x): Multiple reviewers questioned the central premise of treating human-generated text as OOD. They requested both theoretical justification and empirical experiments to validate this assumption.

Authors' Response: The authors addressed this by providing new experiments measuring the intra-distance of LLM/human texts and the inter-distance between them, arguing the results validate their OOD setup. They also added a direct empirical comparison using LLM-generated text as the OOD set, as requested.

2. Experimental Rigor and Ablation Studies
Reviewers' Points (mbi1, vd2x, twCw): Reviewers requested crucial ablation studies to isolate the effects of the different loss components (contrastive vs. binary classification). They also asked for sensitivity analysis regarding training set size and hyperparameters, as well as a comparison of computational efficiency (training/inference time).

Authors' Response: The authors provided the requested ablation studies by testing model configurations with β=0 and α=0. They also added a new sensitivity analysis for training data size and hyperparameters and included a table comparing training/inference times against baselines.

3. Scope of Evaluation and Baselines
Reviewers' Points (twCw, vd2x): Concerns were raised about the limited scope of the evaluation. Reviewers noted a lack of comparison to prior one-class/OOD text detectors, the use of only a single model backbone (RoBERTa), and the need for testing on more recent LLMs.

Authors' Response: The authors clarified the distinction between their research setting and that of prior work. They expanded their evaluation by providing results for more efficient and multilingual backbones. Crucially, they added new results on more challenging and recent datasets, including a newer subset of RAID and a GPT-4o subset of EvoBench.

4. Theoretical and Empirical Divergence Estimation
Reviewer's Point (twCw): One reviewer pointed out the absence of an empirical estimation for the paper's theoretical χ²-divergence claim.

Authors' Response: The authors addressed this by adding an empirical estimation of the χ²-divergence to the paper, along with further analysis to strengthen their theoretical claims.

Therefore the AC recommends acceptance.